# Methane emissions from an oil sands tailings pond: A quantitative comparison of fluxes derived by different methods

Yuan You[1, §], Ralf M. Staebler[1], Samar G. Moussa[1], James Beck[2], Richard L. Mittermeier[1]

[1]Air Quality Research Division, Environment and Climate Change Canada (ECCC), Toronto, M3H 5T4, Canada
[2]Suncor Energy Inc., Calgary, T2P 3Y7, Canada
[§]Now at Department of Physics, University of Toronto, Toronto, M5S 1A7, Canada

*Correspondence:* Ralf M. Staebler (ralf.staebler@canada.ca)

**Abstract.** Tailings ponds in the Alberta Oil Sands Region are significant sources of fugitive emissions of methane to the atmosphere, but detailed knowledge on spatial and temporal variabilities is lacking due to limitations of the methods deployed under current regulatory compliance monitoring programs. To develop more robust and representative methods for quantifying fugitive emissions, three micrometeorological flux methods (eddy covariance, gradient, and inverse dispersion) were applied along with traditional flux chambers to determine fluxes over a 5-week period. Eddy covariance flux measurements provided the benchmark. A method is presented to directly calculate stability-corrected eddy diffusivities that can be applied to vertical gas profiles for gradient flux estimation. Gradient fluxes were shown to agree with eddy covariance within 18%, while inverse dispersion model flux estimates were 30% lower. Fluxes were shown to have only a minor diurnal cycle (15% variability) and were weakly dependent on wind speed, air and water surface temperatures. Flux chambers underestimated the fluxes by 64% in this particular campaign. The results show that the larger footprint together with high temporal resolution of micrometeorological flux measurement methods may result in more robust estimates of the pond greenhouse gas emissions.

## 1 Introduction

Fossil fuel deposits in the Alberta Oil Sands Region - consist of a mixture of quartz sands, slit, clay, bitumen, organics, trace metals, minerals, trapped gases, and pore water (Small et al., 2015). Surface mining is widely practised to extract the oil sands where the deposits are shallow. Extraction of the bitumen from the oil sands involves large amounts of warm water, various additives such as caustic soda and sodium citrate, and diluents, such as naphtha or paraffin (Simpson et al., 2010; Small et al., 2015). Non-recovered diluents, additives, and bitumen, along with water end up in large engineered tailings ponds.

There have been a number of studies to quantify the emissions of pollutants to the atmosphere from the various industrial activities associated with the oil sands (Simpson et al., 2010; Liggio et al., 2016; Li et al., 2017; Liggio et al., 2017; Baray et al., 2018; Liggio et al., 2019). Pollutant emissions that have been observed from tailings ponds include greenhouse gases (GHGs, mainly methane ($CH_4$) and carbon dioxide ($CO_2$)), reduced sulphur compounds, volatile organic compounds (VOCs), and polycyclic aromatic hydrocarbons (PAHs) (Siddique et al., 2007; Simpson et al., 2010; Yeh et al., 2010; Siddique et al., 2011; Siddique et al., 2012; Galarneau et al. 2014; Small et al., 2015; Bari and Kindzierski, 2018; Zhang et al.,2019). However, published studies on atmospheric emissions from tailings

ponds have been rare (Galarneau et al., 2014; Small et al. 2015; Zhang et al. 2019), and significant gaps remain regarding their contribution to total emission from oil sands operations (Small et al., 2015).

Quantifying greenhouse gases emissions from tailings pond is essential, since facilities are required to report specified gas emissions (Government of Alberta, 2019) and to follow emission standards (Statutes of Alberta, 2016). $CH_4$ is long-lived in the atmosphere and has a greenhouse gas global warming potential (GWP) per molecule that is 28 times that of $CO_2$ on a 100-year time horizon, contributing 0.97 W m$^{-2}$ radiative forcing to the total of 2.83 W m$^{-2}$ by all well-mixed greenhouse gases since the beginning of the industrial era (Myhre et al., 2013). $CH_4$ can be produced by microbes in the oil sands tailings through methanogenic degradation of hydrocarbon in diluents and unrecovered bitumen (Siddique et al., 2007; Penner and Foght, 2010; Siddique et al., 2011; Siddique et al., 2012; Foght et al., 2017; Kong et al., 2019).

Most commonly, flux chambers have been used to determine emission rate of GHGs from tailings ponds (Small et al., 2015; Stantec, 2016). These chambers cover an area of less than 1 m$^2$ each and result in only short snapshots of emissions that may not capture the spatiotemporal variability of emissions. Tailings ponds in the oil sands region typically have a size of 0.1-10 km$^2$ with heterogeneous surfaces. Micrometeorological methods of determining fluxes, such as eddy covariance (EC) (Foken et al., 2012) and gradient fluxes (Meyers et al., 1996), are non-intrusive and continuous methods that can be used to measure fluxes from area sources. These methods intrinsically produce integrated flux estimates representative of hectares to km$^2$. In addition, inverse dispersion models (IDMs) (Flesch et al., 1995) and vertical radial plume mapping (VRPM) (Hashmonay et al., 2001) can be used to combine micrometeorological information with measured pollutant concentrations to deduce surface-atmosphere exchange rates.

Micrometeorological methods applied to large areas of a tailings pond can provide much needed information on the spatial and temporal variabilities of emission fluxes from tailings ponds as an input for air quality and climate change modelling. Tailings ponds represent a useful testing ground for a multi-method comparison of flux measurement techniques due to their reliability as sources of significant fluxes, relatively well-defined sources areas, and minimal other anthropogenic sources in the immediate vicinity. This manuscript describes the results of a comparison of flux chambers, EC, gradient and IDM approaches for estimating emission rates of $CH_4$, to verify the suitability of these methods for quantifying fugitive emissions from such sources.

## 2   Site and measurement description

The main site of this study was on the south shore of Suncor Pond 2/3 (Fig. 1; 56°59'0.90"N, 111°30'30.30"W, 305m ASL). The Suncor main facility was 2.6 km to the northeast, and the Syncrude main facility 9 km to the northwest. The pond liquid surface area was about 2.5 km by 1.3 km. Within 2 km to the south of our measurement site, the landscape included natural landscapes, a workers camp, and parking lots. There were also other facilities and sources around the pond, but too far from our measurement site to contribute to the fluxes measured using the methods in this study (Section 4.2). Measurements were conducted from July 28 to September 5, 2017. The sampling platform was a 32 m mobile tower instrumented at three levels (8m, 18m, and 32m) above ground, plus another sampling level at 4 m above ground on the roof of the trailer housing the instruments. This setup allowed the measurement of the vertical

gradient of gaseous pollutants concentrations and meteorological conditions. Gas inlets at these levels were connected to a range of instruments in the trailer located right beside the flux tower, through 40 m of ½" (1.27 cm) outer diameter Teflon tubing for the upper three levels and 7 m of tubing for the lowest level. For the gradient measurements, a cavity ring-down spectroscopy instrument (Picarro, Model G2204) was used to measure $CH_4$ and hydrogen sulfide ($H_2S$) at four levels by cycling through the levels every 10 minutes (i.e. 2.5 minutes at each level). Readings from the first 30

75    s after each level switch were discarded.

For the EC measurements, another cavity ring-down spectroscopy (CRDS) instrument (Picarro, Model G2311f) was used to measure the mole fraction of $CH_4$, $CO_2$ and $H_2O$ (water vapor) at 10Hz. It sampled from the 18m level through a 30 m 3/8" outer diameter Teflon tube at a flow rate of 7 L min$^{-1}$.

Calibrations of $CH_4$ for all the CRDS instruments were performed before and after the field project against secondary

standards traceable to standards used by Environment and Climate Change Canada (ECCC) for their GHG Observational Program, which are in turn traceable to World Meteorological Organization (WMO) standards.

At each of the three levels on the tower, an ultrasonic anemometer (Campbell Scientific, Model CSAT3) measured the turbulent motions in the atmosphere, i.e. u, v, w (the three orthogonal components of the wind) and T (sonic temperature), at 10Hz. The momentum flux and the sensible heat flux can be calculated from the covariance of the

vertical wind component with horizontal wind and temperature fluctuations respectively through EC. Friction velocity ($u_*$) can also be calculated from measured u, v, and w ($u_* = (\overline{u'w'}^2 + \overline{v'w'}^2)^{1/4}$). The two lower ultrasonic anemometers pointed towards true north, whereas the ultrasonic anemometer at 32 m pointed at 3.5°. An adjustment to the true north was applied during analysis. There was also a propeller anemometer (Campbell, Model 05103-10) on the trailer roof 4 m above ground, measuring wind speed and direction. Ambient temperature and relative humidity

(RH) were measured with sensors at three levels on the tower and 1m above ground (Rotronic, Model HC2-S3-L; shield: Campbell Scientific, Model 43502). Ambient pressure was measured with a barometer (RM Young Model 61202). A net radiometer (Kipp & Zonen, Model CNR1) was used to measure solar radiation during the entire project. An infrared remote sensor (Campbell Scientific, Model SI-111) was mounted at 32m on the tower looking down at an angle of 30° below horizontal to measure the temperature at the pond surface. With an angular field of view of 44°,

this results in a footprint ranging from 25 m to 228 m from the tower. Given the location of the tower relative to the pond, winds from between 286° and 76° were defined as coming from the pond (Fig. 1).

An open-path Fourier transform infrared (OP-FTIR) spectrometer system (Open Path Air Monitoring System (OPS), Bruker) was set up at the site to measure line-integrated mole fractions of $CH_4$ and other pollutants. The spectrometer was set up in a trailer next to the main tower about 1.7 m above the ground, pointing to three retro-reflectors 200 m to

the east. The lowest retro-reflector was on a tripod, and the higher two retro-reflectors were supported by JLG basket lifts, resulting in heights of the three retro-reflectors of approximately 1.7 m, 11 m, and 23 m above ground. The spectrometer automatically cycled through pointing at these three sequentially. Spectra were measured at a resolution of 0.5 cm$^{-1}$ with 250 scans co-added, resulting in roughly one-minute resolution. Other details on the OP-FTIR setup and spectral retrieval analysis can be found in You et al. (2020).

## 3    Methods for deriving fluxes

### 3.1   Eddy covariance flux

EC fluxes represent a direct measurement of the turbulent vertical exchange of a substance, and as such usually serves as a reference (Foken et al., 2012) to which more indirect methods (such as those described below) can be compared (Bolinius et al., 2016; Prajapati and Santos, 2018). EC typically requires fast response time measurements (on the order of 0.1 seconds) and high sampling frequency (> 5Hz) (Foken et al., 2012), which in this study limits the method to sensible and latent heat ($H_2O$) fluxes, momentum, $CO_2$ and $CH_4$ fluxes.

As summarized in Foken et al. (2012), in the EC method, flux is calculated by averaging the product of the deviations of the vertical wind component and a mole fraction from their means. For compound $c$ and vertical wind component $w$, the flux $F_c$ is thus

$$F_c(EC) = \overline{w'c'} \tag{1}$$

where the mole fraction $c = \bar{c} + c'$, with the overbar denoting the average and the prime a deviation from it, and similarly for $w$. To account for "storage", i.e. the vertical build-up or venting of a gas between the source and the measurement level (assuming a linear vertical profile of gas concentration), a storage term is added, so that the total flux is given by

$$F_c = \overline{w'c'} + \int_0^z \frac{\partial c}{\partial t} \partial z \tag{2}$$

In this study, 30-min averages of the EC flux of $CH_4$ were calculated by combining the 18 m CRDS $CH_4$ data with the CSAT measurements. The raw data were processed by Eddypro (version 6.0.0, LI-COR Inc.), and major processes included axis rotation (double rotation) (cf. Wilczak, et al., 2001), time lag compensation (covariance maximization method) (Fan, et al., 1990), and storage term correction (Foken et al., 2012). The time lag on average was 10.5 seconds. Covariance spectra were examined for signal losses at higher frequencies (smaller eddies) during transit of the sampled air through the sample line, finite sample cell volume, and instrument response (Fig. S1), accounting for a loss of typically 15% of covariance signal, compared to the sensible heat cospectrum that does not suffer from equivalent losses. Spectral corrections following Horst (1997) were applied to correct for these losses. Corrections for signal losses at the low frequency end of the spectral peak due to the finite averaging time were applied according to Moncrieff et al. (2004). The EC flux quality flag was categorized into 3 classes: 0 (best quality), 1 (good quality), and 2 (poor quality) (Mauder et al., 2006; Mauder and Foken, 2004). Only EC fluxes with flag 0 or 1 were included in further analysis.

Although the slope of the shoreline of the pond was very gentle and the wind was not expected to experience any significant perturbations near the flux tower, we also tested calculating the $CH_4$ EC flux using a sector wise planar-fit coordinate rotation (Wilczak et al.,2001). Four sectors were defined: 286º - 76º (pond sector); 76º - 124º (east shoreline sector), 124º -259º (the south sector); 259º -286º (west shoreline sector). The resulting half-hour $CH_4$ EC flux and the flux using double rotation were within $0.0 \pm 0.1$ g m$^{-2}$ d$^{-1}$ of each other (mean and standard deviation of the difference). Therefore, as expected, during this campaign at this site the planar fitting method did not significantly change the final $CH_4$ EC flux results.

## 3.2 Gradient flux method

Gradient flux estimates are based on relationships between the vertical gradient of mole fractions and the associated flux (down the gradient from high to low mole fractions). In the atmosphere, turbulent exchange dominates molecular diffusion by several orders of magnitude under most conditions, and the factor relating the gradient to the flux is a transfer coefficient dependent on the characteristics of turbulence (first–order closure, $K$-theory), called the eddy diffusivity ($K_c$) (Stull, 2003a). The flux is then given by

$$F_c = -K_c \frac{\partial c}{\partial z} \tag{3}$$

Where $F_c$ is the gradient flux for a pollutant $c$, and $\frac{\partial c}{\partial z}$ is the vertical mole fraction gradient. Note that in this notation, $K_c$ incorporates any stability corrections required since stability effects on the relationship between vertical mole fraction gradients and turbulent fluxes are already incorporated. Our approach follows the well-established "modified Bowen Ratio" (MBR) method (Meyers et al., 1996; Bolinius et al., 2016). To calculate $K_c$ of $CH_4$, the measurements of $CH_4$ EC flux and a gradient of mole fraction are required by Eq. (3). From the measurements at the 18 m, we have a direct EC flux for $CH_4$. Since the footprint of fluxes derived from mole fraction gradients between 8 m and 32 m is approximately equivalent to the EC footprint at 18 m (see discussion in Section 4.2), this gradient can be combined with the EC flux to calculate $K_c$ by Eq. (3). However, only a fraction of the observations yield well-resolved $CH_4$ fluxes and gradients, whereas a continuous time series of $K_m$, the eddy diffusivity for momentum (wind speed) by Eq. (4) (Stull, 2003a), can be readily established. Therefore, we establish a relationship between $K_c$ and $K_m$ for those periods when this is feasible, and calculate the ratio of these two, which by definition is the so-called "Schmidt Number" in Eq. (5) (Gualtieri et al. 2017),

$$F_m = -K_m \frac{\partial u}{\partial z} \tag{4}$$

$$S_c = \frac{K_m}{K_c} \tag{5}$$

To get the Schmidt Number $S_c$ by Eq. (5), two approaches were used: the first approach is with a constant $S_c$. A linear regression of binned $K_c$ versus $K_m$ bins was performed. The inverse of this slope (Fig. 2), as defined in Eq. (5), is the Schmidt Number. The least-squared fit produces a $S_c = 0.74$, which falls between published values of 0.99 by Gualtieri et al. (2017), and the average value of 0.6 in Flesch et al. (2002). Since due to the intermittent nature of $CH_4$ a measured $K_c$ is only available a fraction of the time, we use the more continuous momentum eddy diffusivity $K_m$ divided by $S_c$ as our $K_c$.

The second approach is with variable $S_c$. Gualtieri et al.(2017) reviewed experimental and numerical simulation studies of turbulent Schmidt number in the atmospheric environment, and reported $S_c$ values from 0.1 to 1.3. Flesch et al. (2002) measured turbulent $S_c$ of a pesticide in the atmosphere from soil emissions. Reported $S_c$ in that study varied from 0.17 to 1.34 and showed that this was not solely due to measurement uncertainty. The $S_c$ in this study also varies significantly over time when the wind was from the pond, from 0.04 to 2.90.

To investigate the real variability in $S_c$, $S_c = K_{m\_measured}/K_{c\_measured}$ was plotted against the stability parameter $z/L$ (Stull, 2003b), where $L$ is the Obukhov length, for periods when the wind was from the pond direction (Fig. 3). Figure 3

shows that $S_c$ becomes small as $z/L$ indicates increasingly unstable turbulent mixing, i.e. an increasing importance of convective (sensible heat driven) turbulence, which is not captured by an uncorrected $K_m$, vs. mechanical (momentum driven) turbulence. $S_c$ varies significantly with $z/L$, and is associated with significant noise near neutral stability ($z/L$ close to 0). To avoid introducing large scatter in the $S_c$ correction near neutral stability, $S_c$ is set as 0.74 when $z/L$ is close to 0. To make the correction function continuous, a step-wise definition for $S_c$ is given:

$$S_c = \begin{cases} 0.08 + 3.13 \times 10^{-9} e^{\left(\frac{\frac{z}{L}+19.5}{1.008}\right)}, & \frac{z}{L} < -0.18 \\ 0.74, & \frac{z}{L} \geq -0.18 \end{cases} \tag{6}$$

This $S_c$ of the entire study period and a time series of $K_c = K_m/S_c$ (corresponding to 8m and 32m measurements) were calculated. 3 point median smoothing was performed with the calculated $K_c$ time series before the gradient flux of $CH_4$ was calculated using Eq. (3). To lessen the impact of extreme outliers, the final pond average fluxes reported were based on gradient fluxes between the 2.5[th] and 97.5[th] percentiles. In the Results and Discussion section, gradient fluxes and plots from the variable $S_c$ approach are shown, and results with the constant $S_c$ approach are shown in Table S1 for comparison.

It is possible to calculate $K_c$ values based on $CO_2$, in order to avoid potential circularity arguments when calculating gradient fluxes of $CH_4$ using this approach. However, the $CO_2$ flux signal from this pond was confounded by the strong natural variability of the $CO_2$ background, and the smaller signal-to-noise ratio of the pond $CO_2$ flux compared to the $CH_4$ flux (Fig. S1). Regardless, $K_c$ values based on $CO_2$ were calculated, and found to be noisier but statistically not different from those based on $CH_4$ (t-test p = 0.09, based on fluxes binned into 16 wind direction sectors). It would also be possible to base the calculated $K_c$ values on the sensible heat flux instead of the momentum flux, but due to the absence of significant heat fluxes at night, this would not provide the continuity that the momentum fluxes afford.

### 3.3 Inverse dispersion fluxes

Inverse dispersion models (IDMs) can be used to derive emission rates estimates based on line-integrated or point mole fraction measurements downwind of a defined source. Required inputs include the turbulence statistics between the source and point of observation. Unlike the EC and gradient techniques, IDMs also require an estimation of the background mole fraction of the pollutant upwind of the source. The backward Lagrangian Stochastic (bLS) models are a specific subtype of IDMs. WindTrax 2.0 (Thunder Beach Scientific, http://www.thunderbeachscientific.com (Flesch et al., 1995)) based on a bLS model, is used in this study. The emission rate Q (g m$^{-2}$d$^{-1}$) is calculated through:

$$Q = \frac{(C - C_b)}{\left(\frac{C}{Q}\right)_{sim}} \tag{7}$$

where $C$ [ppm] is the pollutant mole fraction at the measurement location, $C_b$ is the background mole fraction of the pollutant, and $(C/Q)_{sim}$ is the simulated ratio of the pollutant mole fraction at the site to the emission rate from the specified source calculated by the bLS model. In this study, the meteorological condition inputs for bLS model are $u_*$ and $L$ taken from the 30-min averaging calculation of ultrasonic anemometer measurements at 8 m, as well as 30-min average wind directions and ambient temperature directly from the propeller and temperature sensor at 4 m. Periods when $u_* < 0.15$ m/s were disregarded (Flesch et al., 2004). $CH_4$ mole fraction input was taken from the OP-FTIR

measurement which was located 10m to the east of the flux tower. Emission rates are calculated by IDM only when the wind came from the pond, including the sectors centred at 270° and 90°.

### 3.4 Flux chamber measurements

Floating flux chamber measurements of $CH_4$ and $CO_2$ were conducted at 15 spots in and around bubbling zones, including 4 within 500 m of the tower, from Aug 31 to September 2, 2017, by Barr Engineering Co., using compliance monitoring procedures established with guidance from the *Quantification of Area Fugitive Emissions at Oil Sands* issued by Alberta Environment and Parks (AEP 2019). On-site analysis of GHG was performed using U.S. Environmental Protection Agency (USEPA) flux chambers with real-time GHG analyzers (Los Gatos Research, Inc., USA). These flux chamber measurements were conducted during daytime. Key procedural steps include 45 minutes of purging pure nitrogen gas to reach an equilibrium between the flow of the inert carrier gas and the methane evolving from the pond surface, and measurement for a minimum of 30 minutes of with steady-state concentration readings. GHG gases reported from the chamber measurements include $CH_4$, $CO_2$ and $N_2O$ (nitrous oxide). Fluxes were calculated according to the USEPA users guide EPA/600/8-86/008 (USEPA 1986, Equation 3-5):

$$F_{chamb} = \frac{Q \times C}{A} \tag{8}$$

Where Q is the flux chamber sweep air flow rate (L min$^{-1}$), A is the enclosed surface area (m$^2$), and C is measured concentration in ($\mu g$ L$^{-1}$).

### 3.5 Area weight-average of flux

To derive fluxes representing the whole pond, the half-hour fluxes (EC, gradient, and IDM fluxes) are binned by wind direction into 16 sectors. Area weighted averages of fluxes for the pond $F_{pond}$ are then calculated by

$$F_{pond} = \frac{\sum_{sectors} \overline{F(flux, sector)} * Area(sector)}{\sum_{sectors} Area(sector)} \tag{9}$$

The area weighted averages of fluxes results are summarized in Table 1 and serve as the final average fluxes representing the whole pond over the study period.

## 4 Results and Discussion

### 4.1 Meteorological conditions

As shown in the wind rose (Fig. S2), wind coming from the pond occurred only about 22% of the entire measurement period. The dominance of winds from the background directions was known before the study, based on records from monitoring stations in the area, but logistical and access constraints limited us to using the south shore for the setup. There was no significant diurnal variation in wind direction over the entire period. The ambient temperature during the measurement period varied from 7.5 to 31.1 °C, with an average of 17.5 °C (Fig. 4(b)). The mean wind speed measured with the propeller anemometer at 4 m was 3.0 m s$^{-1}$, with a range from 0 to 14.9 m s$^{-1}$ and quartiles of 1.7 and 4.0 m s$^{-1}$ (Fig.4(a)). The mean friction velocity at 8 m (the lowest height by sonic anemometer measurement) over the whole measurement period was 0.32 m s$^{-1}$ (Fig. 4(a)), with a range from 0.03 to 1.01 m s$^{-1}$ and quartiles are 0.20

and 0.42 m s$^{-1}$. Wind speed and friction velocity had a predictable diurnal pattern, greater during the day than at night

(Fig. 4(a)).

In Fort McMurray during the study period, the sunrise was in the range of 4:35 to 5:56 MDT (Mountain Daylight Savings Time, UTC-6), solar noon occurs at around 13:30 and sunset occurs in the range of 22:25 to 20:49 MDT (Fig. 4(d)). Winds across the pond and from the south pass over markedly different surface types (liquid pond vs. a mixture of solid surface types), so the sensible heat flux $H$ is analyzed separately based on the wind direction (Fig. 4(e)).

During the day (from 8:00 to 19:00), $H$ associated with winds across the pond was consistently smaller than $H$ with winds from other directions, suggesting the pond absorbs significant solar energy at the site during the day. It is also worth mentioning that $H$ stayed positive during the night when the wind came across the pond, consistent with the observation that the pond surface temperature was greater than the air temperature (Fig. 4(c)). These resulted in convective turbulent transport of species emitted from the pond surface throughout the night.

**4.2    Footprint of flux measurements**

The footprint of a micrometeorological flux measurement, i.e. the area upwind that contributes to the flux at the point of observation, depends on the wind speed and the dynamic stability of the surface layer. The footprints of EC fluxes measured at 18m at each half-hour period were estimated using the algorithm by Kljun et al. (2015), which takes mean wind speed, boundary layer height, wind direction, friction velocity, Obukhov length, and standard deviation of

horizontal wind speed. Boundary layer height was estimated using the LIDAR measurements at Fort McKay in August 2017 (Strawbridge et al., 2018). Footprints under unstable conditions are summarized in the polar plot in Fig. 1. Footprint contribution distances were calculated for each half-hour over the entire period of study. Results were further separated into unstable ($z/L \leq$ -0.0625), neutral (-0.0625 < $z/L$< 0.0625), and stable ($z/L \geq$ 0.0625) conditions. Since unstable conditions applied 98.6% of time when the wind was from the pond and 52% of entire measurement period,

we summarized the unstable conditions footprint results into 16 wind direction bins, and medians are shown in the polar plot in Fig.1 (footprint under neutral and stable conditions are shown in Fig. S3). The footprint results show the EC flux footprint lies mostly within the edges of the pond.

For gradient flux measurements, the effective footprint is the same as the EC footprint at the geometric mean of the two sample heights (Horst, 1999), for a homogeneous surface area upwind. In this study, gradients between 8 m and

32 m therefore have a footprint equivalent to that for EC at 16 m, reasonably close to where the 18 m EC fluxes were measured.  Since the concentration footprint at the upper (32 m) level is larger than the concentration footprint at the lower (8 m) level, the gradient flux may be affected by sources beyond the geometric mean footprint.

**4.3    Eddy covariance flux**

Analysis of $CH_4$ mole fractions at 18 m as shown in Fig. 5 clearly indicates that $CH_4$ was elevated when the wind was

from the pond direction, and it was steady at round 1.9 ppm when the wind was from other directions (Fig. 5 and 6). Besides sectors from the pond directions, Fig. 7 shows $CH_4$ fluxes significantly larger than zero from two sectors centred with 90° and 270°, i.e. along the shorelines to the east and west. Therefore, measured results for air coming

from these two sectors could represent a mixture of air carrying pond emissions and air from the shore. EC fluxes from the four wind directions sectors centred in the range of 292.5° to 0° are close to each other.

There was no statistically significant diurnal pattern of the $CH_4$ EC flux when the wind came from the pond direction (WD ≥ 286°, or WD ≤ 76°) (relative standard deviation is 15%, p = 0.54) (Fig. S4 (a)). The diurnal pattern of another three sectors when the wind was not from the pond were studied. The sector 259° ≤ WD < 286° (Fig. S4 (b)) contains a mixture of pond emission and the shore of pond, and it also showed no significant diurnal pattern. The sector 214° ≤ WD < 259° (Fig.S4 (c) mainly covers trees and a lake, and showed a slightly increased flux during 12:00-18:00,

which is likely due to biogenic emission from trees and soils (Covey and Megonigal, 2019). The sector 124° ≤ WD < 146° (Fig. S4 (d)) covered a workers' lodge and parking lots, and $CH_4$ emissions and diurnal variation were close to zero. The lack of a diurnal variation of $CH_4$ EC flux observed when the wind was from the pond in this study was similar to the lack of diurnal variation of $CH_4$ EC flux at another tailings pond reported by Zhang et al. (2019). Relationships between the flux when the wind was from the pond and various meteorological parameters were

investigated, and results show that fluxes showed weak dependence on wind speed, $u_*$, water surface temperature, or the temperature difference between the water surface and 8 m (Fig. S5), i.e. they were not major drivers of the $CH_4$ emission rate. $CH_4$ at this site is mainly produced through the methanogenesis of hydrocarbon by the microbes in the fine tailings covering a range of depth in the pond (Penner and Foght, 2010; Siddique et al., 2011;Siddique et al., 2012), and therefore is not directly affected much by the meteorological conditions at the surface or above the pond.

**4.4    $CH_4$ gradient flux and comparison with EC flux**

The $CH_4$ mole fraction measured at 8 m and 32 m show that winds across the pond carried significantly more $CH_4$ than from other directions, and there was a clear vertical gradient with mole fraction at 8 m on the order of 0.5 ppm or more higher than at 32 m (Fig. 6). Gradient fluxes were calculated for all periods when valid EC fluxes and concentration gradients were available. The gradient flux derived from measurements at 8m and 32m shows that the

flux was minimal when the wind was from other directions, similar to the EC flux (Fig. S6). Due to significant scatter, the half-hour gradient fluxes were statistically different from the EC fluxes when the wind was from the pond direction (p=0.003). They were moderately correlated (slope=0.80, r=0.32, Fig. S7(a)). To obtain some comparability, it is therefore necessary to average blocks of data into appropriate bins. A t-test of the gradient and eddy average fluxes binned by wind direction (22.5º blocks) yielded a p = 0.30, and hourly diurnal averaged fluxes agreed with a p= 0.09.

The pond area weighted mean gradient flux was 8% lower than EC flux, and the median was 18% less than EC flux (Table 1).

Studies comparing MBR and EC $CH_4$ fluxes are rare. Zhao et al. (2019) compared $CH_4$ fluxes from an MBR method as well as from an aerodynamic flux model to EC fluxes for two small fish ponds, and showed that the MBR fluxes were well correlated with EC fluxes, with a mean 27% greater than the EC mean flux. The gradient flux calculation

in our study can be considered a hybrid of the MBR and aerodynamic methods, based on a continuous time series of eddy diffusivities for momentum, scaled by the eddy diffusivity for $CH_4$. The gradient fluxes of $CH_4$ agreed well with EC flux in our study, providing a basis for applying the derived $K_c$ values to calculate gradient fluxes for a variety of other gases emitted by the pond (e.g. You et al. (2021)). Other studies comparing MBR with eddy covariance methods

on other gases fluxes, such as $CO_2$, have been reported. Xiao et al. (2014) showed that fluxes of $CO_2$ from these two methods were comparable at Lake Taihu. Wolf et al. (2008) and Bolinius et al. (2016) used EC of heat to derive gradient fluxes of $CO_2$ over trees, and showed they were comparable with EC fluxes.

Gradient fluxes were also calculated with the constant $S_c$ approach, as described in Section 3.2, and results are listed in Table S1. Gradient fluxes calculated from a constant $S_c$ were significantly lower than gradient fluxes with the variable $S_c$ approach ($p < 10^{-21}$, the pond average mean (median) is 33% (34%) lower respectively). Results from this study clearly present the variable nature of $S_c$, and that correcting $S_c$ with stability ($z/L$) is effective to improve gradient flux calculations. While the function derived (Eq. (6)) is primarily a function of the characteristics of atmospheric turbulence and should have broad applicability, it is based on a limited data set and should be verified in other settings in future studies.

### 4.5    $CH_4$ inverse dispersion flux and comparison with EC flux

Compared to point measurements, path-integrated measurements have the advantage of being less sensitive to changes of wind direction and being representative of larger areal averages (Flesch et al. 2004). Therefore, the bottom path-integrated $CH_4$ mole fraction of the FTIR was used as input for IDM flux estimate. The bottom path measurement had the greatest signal-to-noise ratio, and a footprint of on the order of 1-2 km, which is comparable to the footprint of the EC and gradient fluxes (Fig. 1). $CH_4$ IDM flux calculated from the path-integrated mole fraction inputs from OP-FTIR bottom path measurements (when the OP-FTIR path was downwind of the pond) compared well to EC flux, based on the set of simultaneous half-hour periods when both EC and IDM fluxes were available. IDM and EC flux showed reasonable correlation (r=0.62) with a slope of 0.69 (Fig. S7(b)), although the averaged half-hour IDM fluxes are significantly different from EC fluxes ($p < 10^{-4}$). Binning into 16 wind direction sectors similar to described in Section 4.4 yielded agreement at the p=0.08 level. The pond area-weighted mean IDM flux was 30% smaller than EC flux, and the pond area-weighted median IDM flux was also 30% smaller than EC median flux. Some of the differences are likely due to the different footprints of the two measurements. The footprint for turbulent fluxes is smaller than the footprint for concentrations at the same height (Schmid, 1994). The IDM flux showed weak diurnal variations when the wind came from the pond directions (Fig. S8), with smaller fluxes during the day, compared to fluxes at night (p = 0.04), inconsistent with EC and gradient fluxes. As stated in Section 3.3, half-hour periods when $u_* < 0.15$ m/s were excluded in IDM calculation (Flesch at al. 2004). This filtering excluded more nighttime fluxes than daytime fluxes, which caused more limited data in IDM nighttime fluxes and biased the t-test.

Since the background mole fraction input for IDM calculation could affect the flux estimates, two approaches of determing backgound mole fraction of $CH_4$ for model inputs were tested: the daily minimum of $CH_4$ from wind sectors between 180° and 240° of OP-FTIR at our site; the $CH_4$ from another independent OP-FTIR measurement on the north shore of this pond (details are described in You et al. (2021)). Results IDM fluxes with these two background approaches agreed well (You et al. 2021).

### 4.6    Flux chamber measurements

Fluxes from the 15 flux chamber measurements over 3 days in and around the bubbling zones varied from 0.9 to 5.1 $g\ m^{-2}\ d^{-1}$, with an average of 2.8 $g\ m^{-2}\ d^{-1}$ and a median of 2.3 $g\ m^{-2}\ d^{-1}$. The average flux of the five measurements on the last day Sept 2 is 3.6 $g\ m^{-2}\ d^{-1}$, which is the highest amongst the 3 days. The great variation amongst these 15 measurements show the pond was highly heterogeneous in terms of $CH_4$ emissions. The average fluxes from these flux chamber measurements are about half of average fluxes from EC, gradient, and IDM methods. While the flux chamber measurements were deployed over the three days, the wind was from the south, so no simultaneous comparison could be made between flux chamber measurements and micrometeorological methods. However, based on the micrometeorological fluxes spanning more than a month, there is no evidence of day-to-day variability of this magnitude, and we conclude that the mismatch is due to spatial or methodological differences.

Annual compliance flux chamber measurements in 2016 resulted in pond average fluxes of 5.3 $g\ m^{-2}\ d^{-1}$, and 11.1 $g\ m^{-2}\ d^{-1}$ in 2018, despite similar operational parameters in these years as in 2017. We conclude that the underestimate in 2017 is not an indication of a systematic bias of flux chambers, but rather a measure of the uncertainty involved in flux estimates based on snapshot chamber measurements.

A few other studies have also discussed differences between flux chambers and micrometeorological methods (Schubert et al. 2012; Podgrajsek et al. 2014; Erkkilä et al. 2018; Zhang et al. 2019). Zhang et al. (2019) measured $CH_4$ emission from another tailings pond, and reported flux chamber measurements were more than 10 times greater than fluxes from the EC method. They stated that strong eruptions of bubbles could overwhelm the chamber to result in a local underestimation of the flux. On the other hand, the lower EC flux estimate suggests that the area average flux was being overestimated by extrapolation from the chambers, which may have preferentially been located over bubble zones. Their EC fluxes were two orders of magnitude smaller than $CH_4$ flux in this study. Results from this study and Zhang et al. (2019) suggest that average tailings pond $CH_4$ emission extrapolated from a few individual flux chamber measurements may significantly underestimate or overestimate fluxes relative to area-averaging micrometeorological measurements.

This has also been shown over other water surfaces. Podgrajsek et al. (2014) investigated $CH_4$ fluxes at the lake Tamnaren, and reported the fluxes from EC and flux chamber were on the same order of magnitude. They stated due to the non-continuous measurement of flux chambers, some high flux short episodes could be missed. Schubert et al. (2012) measured $CH_4$ fluxes from Lake Rotsee, and reported the fluxes from EC and flux chamber compared well. Erkkilä et al. (2018) measured $CH_4$ flux at Lake Kuivajärvi with the two methods when the lake was stratified, and reported flux chamber measurements were significantly greater than EC fluxes. In conclusion, while flux chambers present advantages in terms of finer spatial and temporal resolution for small sources or locations with high spatial heterogeneity, reliance on a limited number of flux chamber measurements can result in significant year-to-year variability, and spatially integrating methods such as eddy covariance or gradient fluxes will generally provide more representative averages.

**4.7 Comparison with previous results**

Emissions reported in Small et al. (2015) and a Stantec report (2016) (Table 2) represent estimates extrapolated from individual flux chamber measurements, and did not incorporate any seasonal variations for microbial $CH_4$ emissions. Therefore, to compare result of this study to results summarized in Small et al. (2015), we simply used 1 year =365 equal days. Small et al. (2015) showed that $CH_4$ emissions from the same pond were 2.6 g m$^{-2}$ d$^{-1}$ based on the averaging of flux chamber measurements during August to October in 2010 and 2011. A Stantec compliance report (2016) presented flux chamber measurements on this pond with resulting average fluxes of 12.9 and 2.1 g m$^{-2}$ d$^{-1}$ (bubbling and quiescent zones, respectively) in 2013, and 9.6 g m$^{-2}$ d$^{-1}$ and below detection limit respectively in 2014. EC fluxes of $CH_4$ in this study are a factor of 2.8 greater than flux chamber measurements which were taken during the last few days of this project and a factor of 3 greater than emissions reported in Small et al. (2015). However, $CH_4$ fluxes in this study are 19% to 40% smaller than the fluxes from the bubbling zones in 2013 and 2014 (Stantec, 2016). The big differences between flux chamber measurements in the bubbling and quiescent zones may suggest micrometeorological measurements with a bigger footprint will perform better in quantifying emission from the whole pond. It is worth noting that the seasonal variation of fugitive emission from tailings pond is still not well understood, and that different daily emissions are derived from the tabulated annual results from Small et al. (2015) depending on the annual extrapolation model used. This reflects a general complication when comparing the five weeks emission results in this study to annual emissions reported in the past.

Baray et al. (2018) calculated $CH_4$ emission from this pond based on airborne measurement in 2013 over the whole facility, combined with reported statistics stating that 58% of $CH_4$ emissions within the facility were from tailings ponds, and 85% of emissions from these tailings ponds were from Pond 2/3. This resulted in an estimate of $2.0 \pm 0.3$ tonnes h$^{-1}$, which converts to 17.1 g m$^{-2}$ d$^{-1}$ (for a pond area of 2.8 km$^2$, Small et al. (2015) Table 2). This emission rate is significantly (119%) greater than emissions from the three micrometeorological methods in this study, possibly because of the uncertainties in the reported percentage contribution of $CH_4$ emissions from this pond to the whole facility.

Suncor reported facility-wide emissions of $CH_4$ for 2017 of 5977 tonnes (Government of Canada, 2017). The emissions measured during the 5 weeks of this study extrapolated to the year result in 6548 tonnes yr$^{-1}$, i.e. 110% of this total. This extrapolation assumes seasonal invariance of $CH_4$ emissions (e.g. January emissions are the same as August emissions) as is common practice in monitoring reports (cf. Stantec, 2016).

When comparing $CH_4$ emissions in this study to emissions summarized in Table 2, it must be kept in mind that different time periods are being compared, and that several factors may contribute to variability of the emissions (Siddique et al. 2007 and 2012). Pond 2/3 is an active pond, and the amount and characteristics of input streams are variable with time. Some of the facility specific variables which could affect the methane emissions include: the amount of diluent loss to the pond, the proportions of diluent and hydrocarbons in the froth treatment tailings (FTT) that enter matured fine tailings (MFT) layers in the pond, density of microbes in the MFT, physical disturbance of the MFT layers, transferring activities of the MFT, pond water temperature change and consequential density inversion between oil layers and water in the pond, FTT discharge diluent composition change, introduction of new materials

and chemicals into the MFT, and consequential change in microbial community (Small et al. 2015; Foght et al. 2017). Natural lakes and wetlands emit at rates typically on the order of 0.005-0.05 g m$^{-2}$ d$^{-1}$ (Sanches et al., 2019). Another independent approach of estimating $CH_4$ emissions is using an emission factor (EF) combined with diluent discharge rates to the pond. The EF was based on an MFT characterization and kinetics of methanogenesis for a matured pond. Pond 2/3 is presumably similar in maturity and properties with the studied MFT in other oil sands facility (Siddique et al. 2008). After incorporating the diluent loss to the pond, the daily $CH_4$ emissions were calculated and integrated into an annual emission of 5860 tonnes, comparable to annual emissions extrapolated from the micrometeorological methods in this study. This approach requires some assumptions: first, that the kinetics of methanogenesis are a function of the maturity of the microbial community within the target MFT; and second, that the properties of the diluent feed stream remain constant over the period considered. This approach can provide emission estimates continually provided that the microbial state in the MFT and the diluent discharge volumes and properties are tracked and remain consistent.

To put the $CH_4$ emissions into a global warming context, the $CH_4$ fluxes can be combined with concurrent flux measurements of $CO_2$ with the same instrumentation. Assuming a GWP of $CH_4$ = 28 (Myhre et al., 2013), the equivalent $CO_2$ flux ($F\_CO_{2eq}$) from this tailings pond $F\_CO_{2eq}=F\_CO_2+ (F\_CH_4\times GWP) = 204$ kilo tonnes year$^{-1}$, 90% of which is due to $CH_4$. This accounts for only 3% of Suncor's facility $CO_{2eq}$ emissions in 2017 due to the dominance of other $CO_2$ sources.

## 5. Conclusions

Results in this study have provided several estimates of the emission of $CH_4$ from this tailings pond using EC, gradient, and IDM, for a period of five weeks. The gradient and inverse dispersion methods agreed moderately with EC results (18% and 30% lower, respectively), which lends confidence that the former two methods can provide valid flux estimates for other gases emanating from the pond. These results were also compared to flux chamber measurements at this pond taken during the study, showing flux chamber estimates were 64% lower than those from micrometeorological methods. The better agreement between the three micrometeorological measurements flux results suggests that the larger footprint of micrometeorological measurements results in more robust emission estimates representing most of the pond area. Fluxes were shown to have only a minor diurnal cycle, with a 15% variability, during the period of this study. To investigate seasonal patterns, further studies measuring $CH_4$ fluxes using micrometeorological methods at this pond or other tailings ponds throughout the year are recommended.

**Data availability.**

All data are publicly available at http://data.ec.gc.ca/data/air/monitor/source-emissions-monitoring-oil-sands-region/emissions-from-tailings-ponds-to-the-atmosphere-oil-sands-region/.

**Author contribution.**

YY and RS conducted the research and wrote the manuscript; SGM contributed flux analysis and editing; RM contributed $CH_4$ data; JB contributed operational data on the pond and contributed to the writing.

**Competing interests.**

Dr. Beck is an employee of Suncor Energy. The other authors have no competing interests.

**Acknowledgements.**

The authors thank the technical team of Andrew Sheppard, Roman Tiuliugenev, Raymon Atienza and Raj Santhaneswaran for their invaluable contributions throughout, Julie Narayan for spatial analysis, Stewart Cober for management and Stoyka Netcheva for home base logistical support. We thank Suncor and its project team (Dan Burt et. al.), AECOM (April Kliachik, Peter Tkalec) and SGS (Nathan Grey, Ardan Ross) for site logistics support.

**Financial support.**

This work was partially funded under the Oil Sands Monitoring Program and is a contribution to the Program but does not necessarily reflect the position of the Program. We also acknowledge funding from the Program for Energy Research and Development (NRCan) and from the Climate Change and Air Quality Program (ECCC).

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

**Tables**

**Table 1 Summary of $CH_4$ fluxes (g m$^{-2}$ d$^{-1}$) in this study.**

| Flux method | Q_25% | median | Q_75% | mean[c] |
|---|---|---|---|---|
| EC[a] | 5.6 | 7.4 | 9.8 | 7.8 ± 1.1 |
| Gradient[a] | 3.8 | 6.1 | 11.0 | 7.2 ± 3.5 |
| IDM[a] | 3.6 | 5.2 | 6.6 | 5.4 ± 0.4 |
| Flux Chamber[b] | 2.0 | 2.3 | 3.8 | 2.8 ± 1.4 |

[a] Statistics and average fluxes are area weight-averaged.
[b] Statistics and average of 15 measurements described in Section 4.6. The error of the mean is the standard deviation of the 15 measurements. Emission estimates were 5.3 g m$^{-2}$ d$^{-1}$ in 2016 and 11.1 g m$^{-2}$ d$^{-1}$ in 2018.
[c] Errors with the mean fluxes are calculated with a "top-down" error estimation approach, using the average of standard deviations of fluxes from five periods when the fluxes displayed high stationarity.

**Table 2 Comparison of $CH_4$ fluxes (g m$^{-2}$ d$^{-1}$) in this study to previously reported fluxes.**

| | TAPOS(2017) | Small et al. (2015)[a] | Stantec report (2016) | | | Baray et al. (2018)[b] | Flux Chamber 2017 |
|---|---|---|---|---|---|---|---|
| | | | | bubbling zones | quiescent zones | | |
| $CH_4$ | 7.8 (EC) | 2.6 | 2013 | 12.9 | 2.1 | 17.1 | 2.8 |
| | | | 2014 | 9.6 | BDL | | |
| $CO_2$ | 24.4 (EC) | 16.4 | 2013 | 14.9 | 10.5 | NA | 21.2 |
| | | | 2014 | 11.0 | BDL | | |

[a] The original units are tonnes hectare$^{-1}$ year$^{-1}$. Measurements were taken from August to October in 2010 or 2011. The pond area was 2.8 km$^2$ as listed in Table 1 of Small et al. (2015). We assumed no seasonal variations to extrapolate from summer measurements to annual totals.
[b] The original number is 2.0 tonnes hour$^{-1}$, and the pond water surface area used was 2.8 km$^2$ (Small et al, 2015).
BDL: below detection limit.
NA: not available.

**Figures**

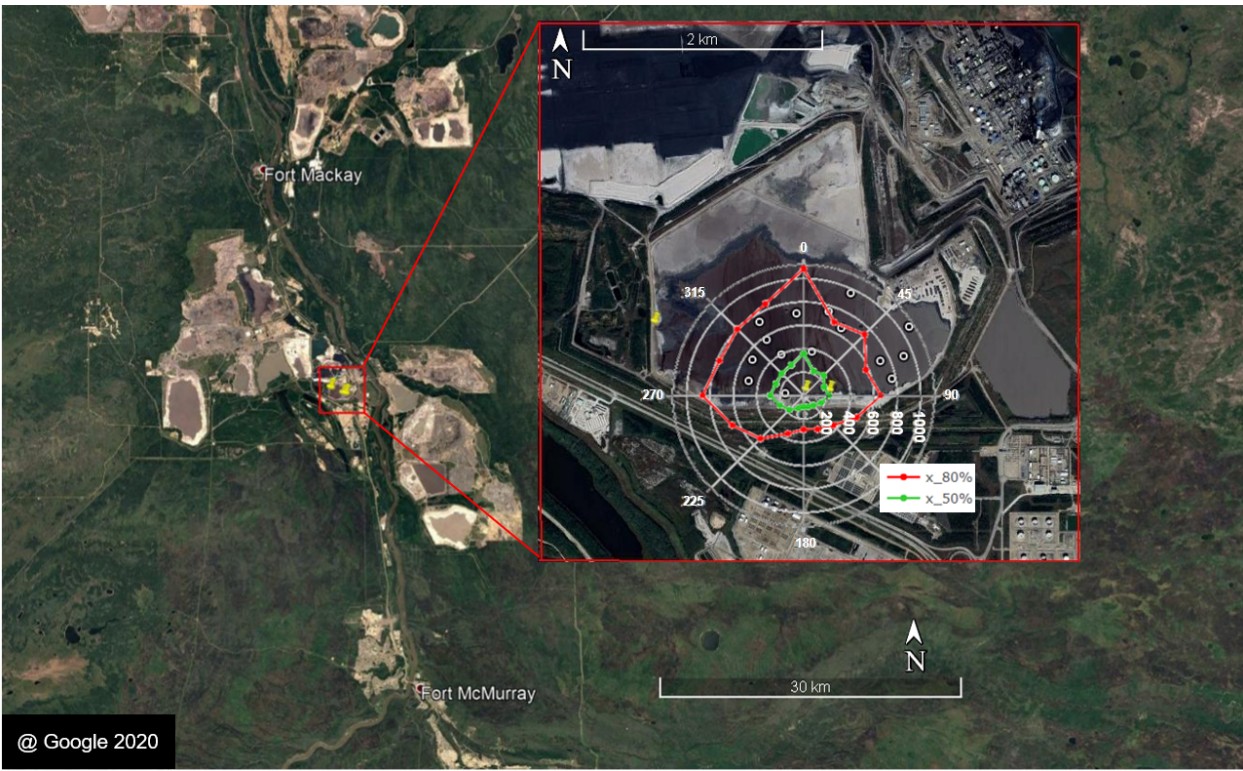

**Figure 1: Overall map of the study site and close-up of the pond in September 2017. The superimposed polar plot shows the footprints under unstable conditions. Two traces in the polar plot show the medians of 80% and 50% contribution distances (in meters) for the measured half-hour periods EC fluxes in 16 wind direction bins. Angles in the polar plot are the wind direction (true North) with the center at the main site. The 15 white circles on the surface of the pond indicate the locations of the flux chamber measurements. The grey areas north of the r=1100m circle are sandy deposits; dark grey represents liquid surfaces.**

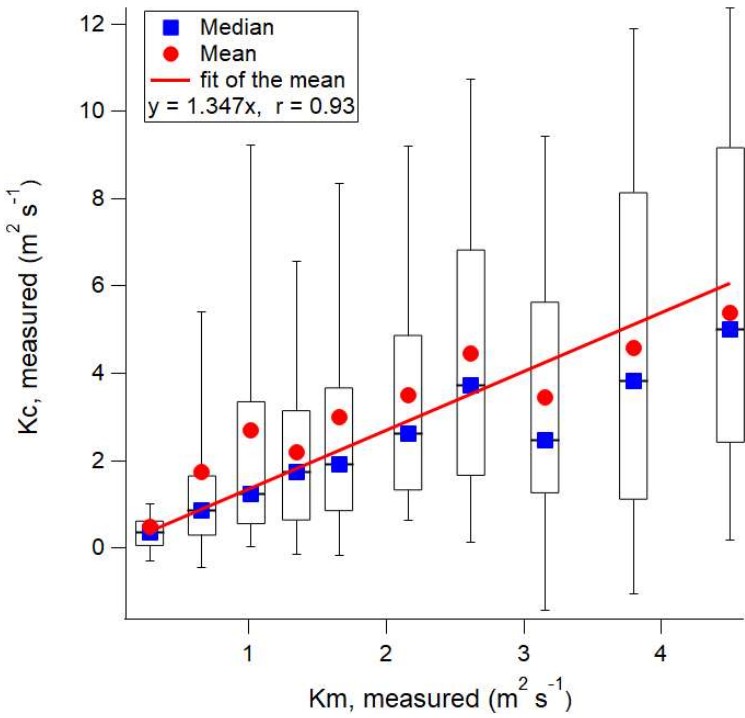

**Figure 2: Calculating Schmidt Number $S_c$ as a constant over the entire study. Lower and upper bounds of the box are 25th and 75th percentile of each bin; the lines in the box and the blue squares mark the median; red circle labels the mean of data in each bin; whiskers are 10th and 90th percentile of data. In this analysis, measured $K_c$ values were binned by $K_m$ with 65 points in each bin. Bin centers were the median $K_m$ measured of each bin. The red line is the best fit of mean $K_c$ vs median $K_m$ of each bin. p value of the fit = 0.0001. Points with $K_m$ > 5 m²s⁻¹ were excluded in the fit.**

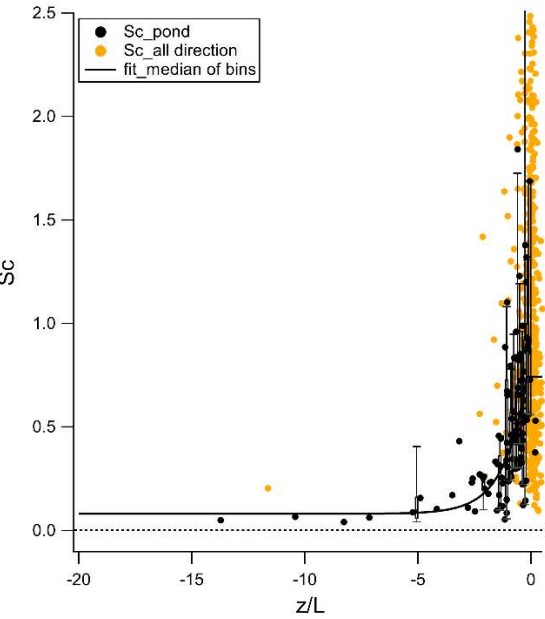

**Figure 3: Dependence of $S_c$ on $z/L$ measured at 18m. Yellow points are $S_c$ observed in each individual half-hour period over the entire period; black points are $S_c$ observed in each individual half-hour period when the wind was from the pond; the black curve is the best fit of $S_c$ verses median $z/L$ from each $z/L$ bin when the wind was from the pond. In this analysis, $S_c$ was binned by $z/L$ with 10 points in each bin before fitting.**

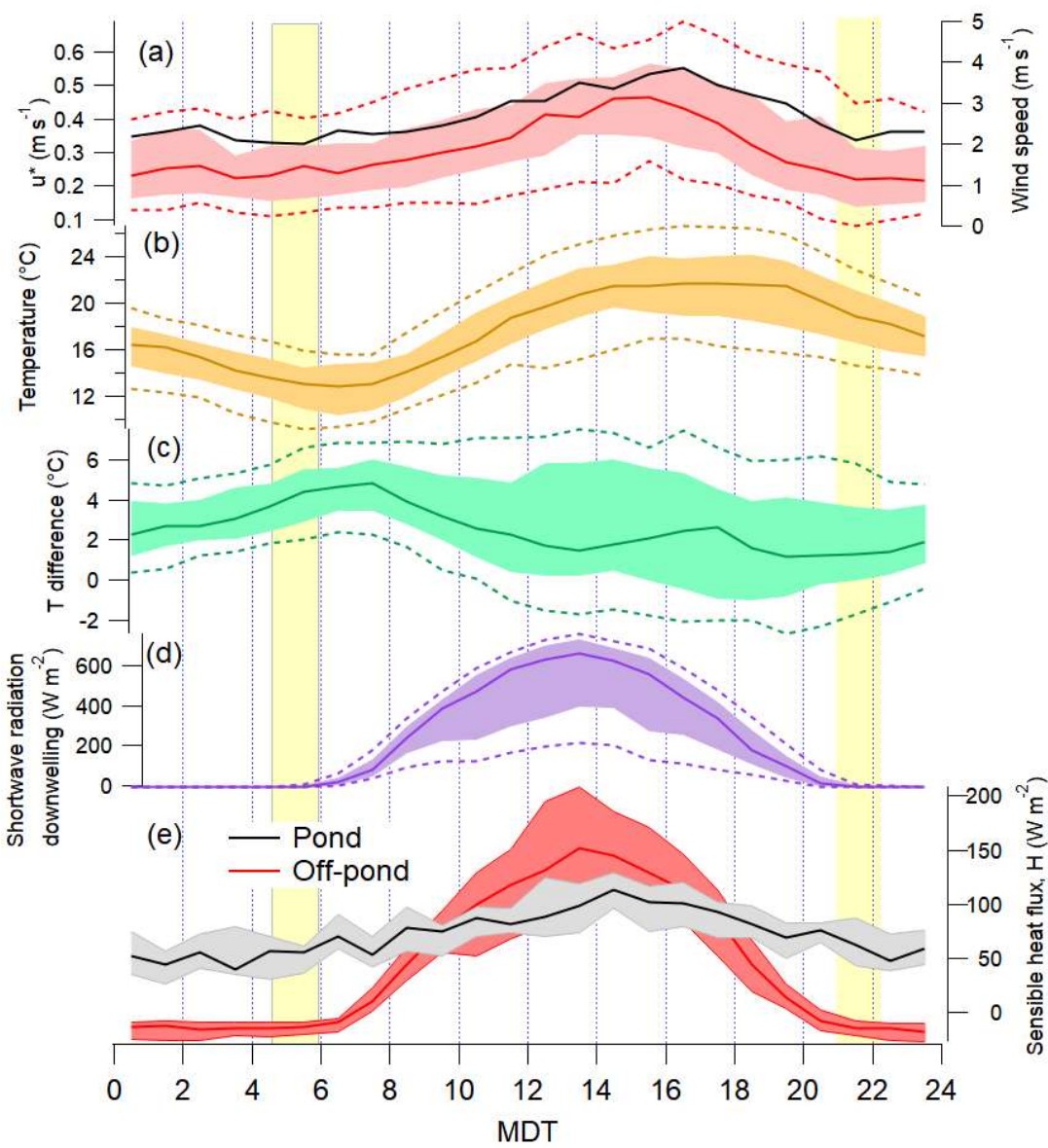

**Figure 4: Diurnal variations of (a) u∗ at 8m (red) and wind speed at 4m (black); (b) ambient temperature at 8 m; (c) the temperature difference between the surface of the pond and the ambient temperature at 8m; (d) downwelling shortwave radiation; (e) the sensible heat flux at 8m. Solid lines show the median; shades indicate the interquartile ranges; and dashed lines label 10th and 90th percentiles. MDT denotes Mountain Daylight Savings Time (hours). The yellow shades mark the range of local sunrise and sunset time during this 5-week project.**

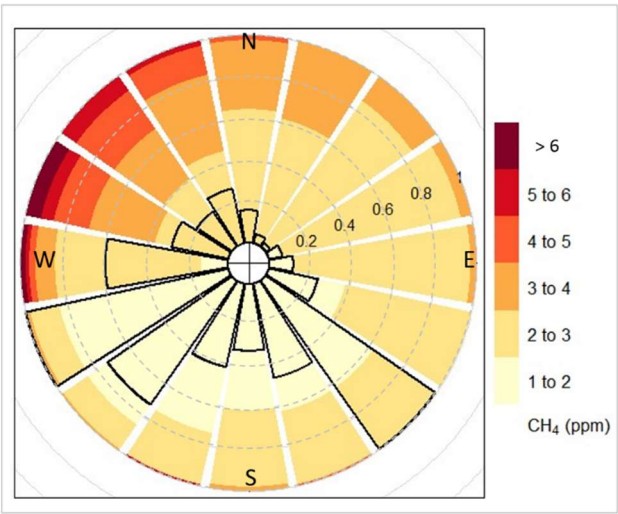

**Figure 5: Rose plot of CH₄ mole fraction at 18m. Colors represent CH₄ mole fraction. The length of each colored segment represents the time fraction of that mole fraction range in each direction bin. The radius of the black open sectors indicates the frequency of wind in each direction bin; angle represents wind direction.**

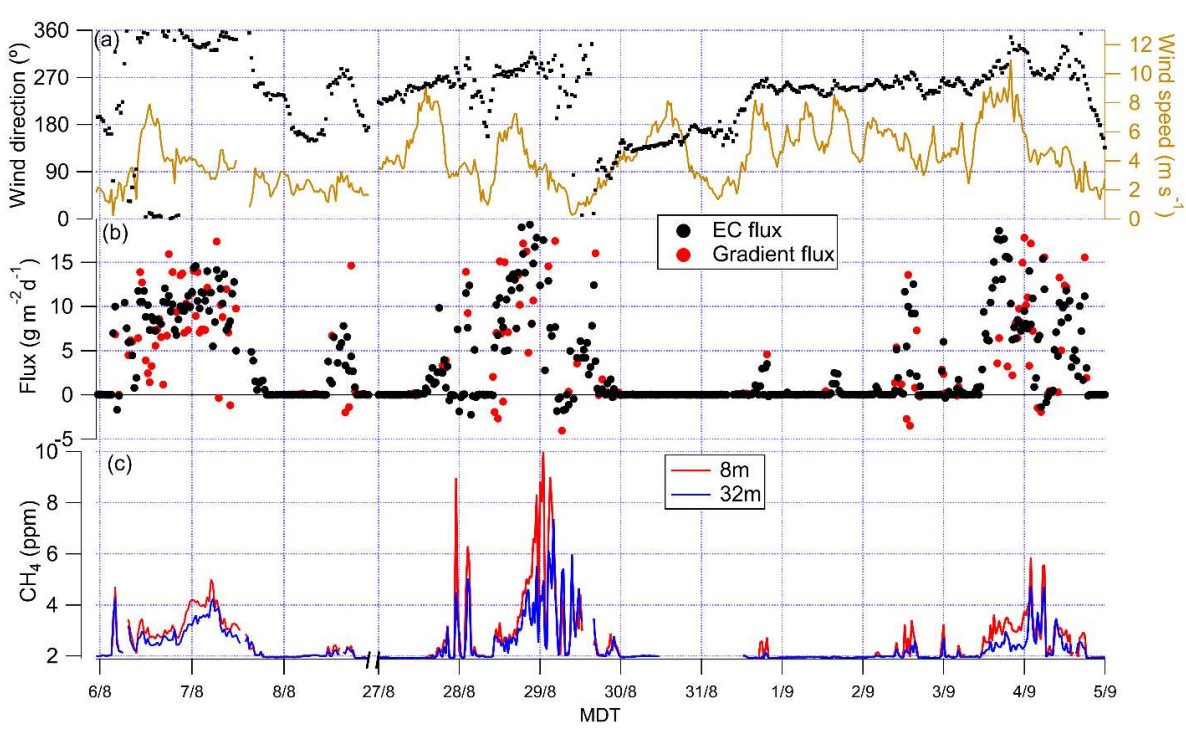

**Figure 6: Time series of (a) wind direction, wind speed, (b) CH₄ EC fluxes and gradient fluxes, and (c) CH₄ mole fractions at 8m and 32m, from Aug 6ᵗʰ to 9ᵗʰ, and from Aug 27 to Sep 5.**

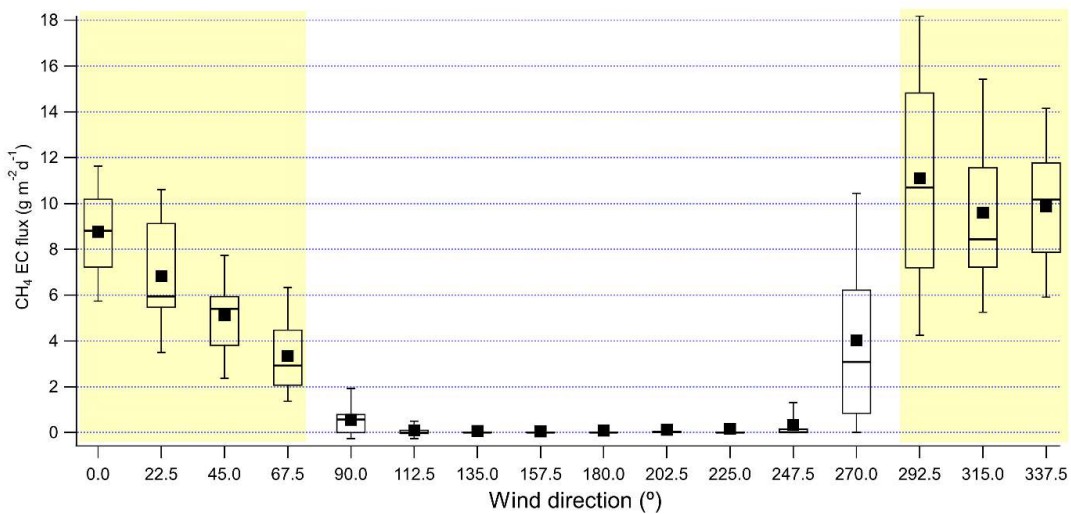

**Figure 7: EC flux of CH$_4$ as a function of wind direction binned in 22.5-degree bins. Lower and upper bounds of the box plot are 25$^{th}$ and 75$^{th}$ percentile; the line in the box marks the median and the black square labels the mean; the whiskers label the 10$^{th}$ and 90$^{th}$ percentile. Yellow shades indicate the wind directions from the pond.**