# Peer review of "Methane emissions from an oil sands tailings pond: A quantitative comparison of fluxes derived by different methods"

_Atmospheric Measurement Techniques, 2020_

## Referee Comment (RC1) · Anonymous Referee #1 · 5 Aug 2020

The manuscript "Methane emissions from an oil sands tailings pond: A quantitative comparison of fluxes derived by different methods" by Yuan You et al. presents a comparison of different methods to estimate Methane (CH4) emissions above trailing ponds in the Albert Oil Sand Region. The described methods including eddy covariance flux measurements, flux estimation from gradient measurements, model estimates based on line-integrated mole fractions measurements as well as flux chamber measurements. The aim of the study is to improve the robustness and representativeness of the used methods to routinely quantifying emissions from fossil fuel deposits. Given the global significance of accurate estimation of CH4 emissions and the lack of studies from waste products of fossil fuel industry the manuscript touches a significant

topic. The used methods and calculations are described in a comprehensive way. All used instrumentation and calibration procedures reflect the current state of research. By using more direct estimates of CH4 flux in order to validate the traditionally used flux chamber measurements the authors reveal a bias in those estimations, caused by the more local deployment of the chambers. Further the authors put their work in the brought context of CH4 measurements done over water bodies as well as GHG estimations from trailing ponds. The manuscript is comprehensive and well written all reached conclusions from the comparison are easy to follow. Besides some minor points that need to be addressed I recommend publication of the manuscript.

Detailed comments directed to the authors:

P 4 LL 119-122: You describe that a standard axis rotation was performed within Eddypro. Could you elaborate a bit more on how this rotation was performed? The abrupt terrain change can pose a problem for measurements obtained at an EC station set up at a shoreline. Especially for the wind sectors that might have contributions from land and water surface. Paw et al. (2000) and Finnigan et al. (2003) suggest considering such terrain structures in the rotation procedure of the eddy-covariance data, which can be obtained by a sector wise application of the planar-fit method according to Wilczak et al. (2001).

Section 3.2: Are there any influences of waves to be expected on the calculation of the gradient fluxes?

Section 4.2: Looking at the wind rose in comparison to the footprint calculation I would expect a more detailed description. Figure S1 indicates the main wind direction from the land side. Further it is stated that only 22% of the half hour fluxes originate from the pond. The shown footprint in Figure S3 suggest a quite large contribution from the pond though. Usually a large footprint extend is connected to stable stratifications however in the your response to the editor you mention that after selection of the wind sector representing the pond hardly any half hour periods with stable conditions were found.

Could you please clarify how the shown footprint fits to the flux data set? Particularly I would find it interesting to see a separation of the footprint for the overall data set as well as unstable, stable and neutral conditions.

In general an overlay of the entire footprint map over a land use map/aerial photo could provide a more useful inside to interpret the data. You mentioned that one reason for the differences between chamber and EC flux calculations, is the local deployment of the chambers. One further approach to gain more information during a comparison of is to use the Kljun model to calculate the land use contribution for each half hour EC flux. This could help to understand the influence of the mentioned bubbling areas on the flux estimates.

Figure S2: In my opinion it does not add much extra information since there is no clear daily pattern. Maybe a marking which direction represents the pond and land sectors would help.

Finnigan J, Clement R, Malhi Y, Leuning R, Cleugh H (2003) A reevaluation of long-term flux measurement techniques, part I: averaging and coordinate rotation. Bound-Lay Meteorol 107:1–48. doi:10.1023/A:1021554900225

Paw UK, Baldocchi D, Meyers T,Wilson K (2000) Correction of eddycovariance measurements incorporating both advective effects and density fluxes. Bound-Lay Meteorol 97:487–511. doi:10.1023/A:1002786702909

Wilczak J, Oncley S, Stage S (2001) Sonic anemometer tilt correction algorithms. Bound-Lay Meteorol 99:127–150. doi:10.1023/A:1018966204465
* * *

---

## Referee Comment (RC2) · Kukka-Maaria Kohonen (Referee) · 22 Sep 2020

Article "Methane emissions from an oil sands tailings pond: A quantitative comparison of fluxes derived by different methods" by You et al. introduces a flux measurement method comparison from methane flux measurements over an oil sands tailings pond. Their analysis includes three micrometeorological flux measurements methods including eddy covariance, floating chamber and gradient flux measurements. In addition, they include in their analysis another method called inversed dispersion fluxes. High methane emissions of the tailings pond are interesting by themselves and allow for a proper flux measurement method comparison, making a good addition to the existing

literature of flux measurement method comparions studies over waterbodies. However, the methods and analysis used in the study require recalculations and corrections for drawing proper conclusions, than what has been done in the present form of the manuscript. The overall presentation of the manuscript is not all the time well structured, and for example the analysis is jumping from only the pond area measurements to fluxes from all different wind directions. Thus, I cannot recommend publication before major revisions are made to the analysis.

General comments:

Measurement methods in general require more description:

- Eddy covariance flux calculation description is lacking relevant information. The authors list as correction methods axis rotation, time lag compensation, WPL correction and storage term correction. Which axis rotation method was used? WPL correction should actually not be applied for this gas analyzer (Picarro G2311-f) as it is already included in the instrument itself. Spectral corrections are not mentioned in the text. Spectral corrections (especially high frequency spectral correction) are essential in EC flux processing and can affect even the sign (direction) of the flux measurement. Recommended spectral correction methods are introduced in e.g. Aubinet et al., (2000) and Mammarella et al., (2009). Was $u^*$ filtering applied? If yes, what was the threshold and how was it determined? How about storage change fluxes, how were they calculated? Fluxes from different wind directions are presented in this study, but it is not clear wether all these fluxes were processed in similar way. If all wind sectors are covered with different types of roughness elements (such as pond, buildings, trees), the different sectors should be processed (and fluxes calculated) individually. Environmental data required for the flux calculation (air pressure, temperature and humidity) are not described.

- Gradient flux method has defeciencies. Eddy diffusivity is calculated from $CH_4$ EC flux, so gradient flux is not totally independent from EC $CH_4$ flux measurements. I

understand the eddy diffusivity is not taken directly based on EC measurements, but from a fit of Schmidt number against stability parameter. Even though making this fit makes gradient fluxes not directly dependent from EC, it should still be discussed how the usage of EC measurements in eddy diffusivity calculations affect the comparison between these methods, as it has not been currently discussed at all. The authors refer to a study by Bolinius et al., 2016 where the eddy diffusivity is calculated from the heat flux measurements of the EC system instead of the gas flux. This is a well established method and I recommend the authors to study it more carefully and implement in their study as well. I suggest the authors at least compare this method to their original gradient flux calculations. Another study worth taking a look at is Rantala et al., (2014), where eddy diffusivity is calculated from the Monin-Obukhov similarity theory. Heat flux is independent from the gas flux, so calculating eddy diffusivity from the heat flux measurements will allow more reliable comparison between EC and gradient CH4 fluxes.

- Chamber measurements are currently not described at all but a proper method description is needed (what kind of chamber design was used, dimensions, how long enclosure time was after reaching equilibrium with carrier gas flow and inside air, how was the air flow implemented, how was the flux calculated, what kind of data selection methods were used etc.).

Comparison of fluxes is highly misleading and fundamentally flawed. The authors have included in the flux averages all data available, which are then compared with each other. What the authors should do instead is to select only those timeperiods/data points when all the measured fluxes are available, and then calculate averages that are comparable. If this is not done, it easily happens so that one of the methods is measuring e.g. more fluxes from one wind direction or time of the day than the other, which is causing a clear bias in the comparison.

Conclusion section is currently an additional discussion section that should have been implemented in the section "Results and discussion" already. Proper conclusions - with

no new information given but rather a summary with a perspective to future studies - is totally missing and should be included.

Specific comments:

Table 1: This comparison does not make sense if the fluxes are not averaged from simultaneous measurements. You should only include the datapoints in averaging when you have a datapoint from all the methods. What does it mean that fluxes are "relatively steady"? The uncertainty estimation in footnote c is unclear.

Table 2: Not clear what are the time periods for these flux estimates, should they even be comparable? Annual averages are different from summertime measurements. It would be interesting to see a comparison to natural waters or reservoirs as well, to see the high magnitude of the methane emissions.

Figure 1: It would be very helpful for the reader to include in the map the EC footprint lines and/or lines for approved wind directions. It is not very clear from the closeup image where exactly are the pond edges. Maybe this could be highlighted somehow? Add chamber measurement locations to this map.

Figure 2: What is the correlation coefficient of the linear fit? How does it change if you use the original datapoints instead of binned averages for fitting? It does seem that data are very scattered with higher Km and Kc, how does this affect the fitting? What do the boxplots represent (what are the box limits, whiskers, center line etc)?

Figure 3: "Best fit" - determined by what criterion? The bins are not of equal size and I believe this is also affecting the fit. What do the boxplots represent (what are the box limits, whiskers, center line etc)?

Figure 4: Fig 4b is not discussed anywhere and is a bit pointless without water temperature. In 4a, u* is missing interquartile ranges and 10% and 90% percentiles. In 4e sensible heat flux is missing quartiles off-pond and 10% and 90% percentiles on pond. Mark in the diurnal plots the times of sunset and sunrise to help the reader.

[Figure]

Figure 5: Scale seems quite arbitrary, how was it defined? Directions are missing, where is north?

Figure 6: You should add a, b and c to subplots. Colors of EC and gradient fluxes are too similar in the printed version and in the lowest panel red and green are used which is not color-blind friendly. You can check colorblind and printer friendly color choises e.g. from here: https://colorbrewer2.org

Figure 7: Shade the pond area also here, similar to Fig. S6. What do the boxplots represent (what are the box limits, whiskers, center line etc)?

Figure 8: What is the offset of the fit? It does not seem to be crossing y=0 at x=0 in neither of the plots.

Figure S2: What do the confidence intervals represent?

Figure S3: "...countours of the EC footprint area". It would be very helpful for the reader to get S3 b on top of a map, to see where the contours are crossing pond edges.

Figure S4: It is not mentioned here which EC flux this is. Methane? Mention in each subplot which wind direction it is representing (in legend/title/xlabel/ylabel) to help the reader. What do the boxplots represent (what are the box limits, whiskers, center line etc)? Mention in the caption what is in each wind sector (pond, buildings, trees, etc).

Figure S5: Mention in the ylabel that this is methane flux. Mention in the caption what is the r2 representing (least squares linear fit?).

Figure S6: What do the boxplots represent (what are the box limits, whiskers, center line etc)?

Figure S7: What do the boxplots represent (what are the box limits, whiskers, center line etc)?

Table S1: Are the fluxes compared here from exact same time periods? Same comment as for Table 1 about the uncertainty estimate and "relatively steady".

L10: "develop" is a little bit misleading here since the authors don't really develop any new method, rather compare already existing ones.

L11-12: Mention briefly which are these three flux methods in one sentence.

L15: inverse dispersion model comes here from out of the blue. Describe it briefly before writing about the results.

L18-19: This sentence is a bit misleading. In one perspective it is quite obvious that a larger footprint represents a larger area. On the other hand if the EC tower is placed so that it is measuring only e.g. shallow area while actually the pond is deeper from a much larger area, then would EC be representative of the whole pond emissions? Then on the other hand nobody can know what is the real flux. It might as well be closer to the chamber flux than EC.

L21: Abbreviation AOSR is not used anywhere in the text

L23: "Oil Sands" or "oil sands"? Throughout the manuscript.

L48: "eddy covariance (EC)" and then use EC after this throughout the manuscript instead of eddy covariance

L49: "area sources" or "source areas"?

L53: So only emissions can be measured with this method, not uptake?

L56: What is meant by "relatively well-defined spatially"? If the fluxes are well-defined, why do you measure them?

L59: "Field study" is not a very descriptive title. Maybe "Site and measurement description"?

L62: Trees are not part of natural landscapes? What is? How far were the other facilities? In the catchment area or further away? How large is the cathment area?

L65: What is meant by "mobile tower"? How high were the measurements above water

(which is more relevant than ground in the case of pond fluxes)?

L69: Is this the diameter or radius? Inner or outer diameter?

L71-72: I am not sure it can be said that turbulent flow is ensured. Reynolds number is ∼1300 according to my calculations, so it is possible that the flow is turbulent, but I wouldn't call it "ensured".

L72-74: All kinds of measurements are presented that are not used in the analysis or shown anywhere in the manuscript. I suggest to leave out the description of those gas measurements not used in this particular study. Why is a 40 m long tubing required for 18 m height measurements? This will cause quite long lag time for EC. What are the three and four levels mentioned here?

L74: There must be some flush time of the tubings and analyzer between the different height measurements. How long is the flush time? One level cannot be measured 2.5 min during 10 min period if you take into account the flush time.

L76-77: Was there any drift of the instruments between calibrations? Did they compare well with each other?

L80-82: This is well known EC theory and does not need to be explained.

L88-89: Was the infrared sensor calibrated somehow?

L92: How were the suitable wind directions determined?

L104: EC also has its limitations, "benchmark" seems a bit exaggerated

L106: Response time and sampling frequency are not the same. Response time should be given in seconds, sampling frequency in hertz. EC measurements require both fast response times and high sampling frequency.

L107: "CO2 and CH4 fluxes"

L108-109: Reformulate the sentence. EC does not calculate anything, and in this case

you are talking about gas fluxes explicitly (not e.g. heat flux since you mention mole fraction)

L113-114: Repetition from above

L115: "storage change flux". Out of curiosity, how large was the storage change flux? Often in lake studies they have been neglected but might be important as well.

L118-122: More description is needed on the processing methods used. How long was the lag time on average?

L122: What do the different flags mean (what are the criterion)?

L123: "Gradient flux method"

L130: units?

L135: How do you define the gradient method footprint?

L168: Shifting winds are also a problem for EC measurements!

L177-180: What are the units of these variables?

L181: Why are L and u* used as inputs, since u* is already used in calculating L?

L185: How close is "right beside"?

L188-195: More description needed

L196-199: Not understood where this is used (which methods) and why. More description please.

L202: "wind coming from" or "wind that came from"

L214: Sunsets and sunrises are not directly seen from Fig 4d.

L215-216: The same wind direction/ source area applies to gas fluxes. Why different wind direction analysis is applied to sensible heat fluxes and not the others?

[Figure]

L229: You can only see this from Fig 4c. What is meant by "species"? Methane?

L225-230: This is done for wind direction filtered data I assume? Should be clarified since all wind directions are analyzed in some way or another in this manuscript.

L230: This is not seen from Fig S3 (b), since you cannot see pond edges in the figure.

L231: "gradient flux"

L234: delete "however"

L235-240: But there is much more data from off-pond direction than from pond direction. How does influence the analysis?

L245-253: It is not clear why fluxes off-pond are reported since this is a study concentrated on pond emissions. Are these sectors processed in flux calculation individually or not?

L253-258: Are these now results from pond direction? Wind and turbulence are still driving the turbulent/diffusive transport of gases from pond to the atmosphere (e.g. Tedford et al., 2014).

L263-265: How do medians correlate? Take into account my earlier comments about representativeness as well.

L270-272: This is quite far taken conclusion. Based on the results here you can only say that EC fluxes were used to calculate Kc, which of course then correlates well with EC...

L272-274: And what were the outcomes of these studies? How do they compare to this study?

L278-281: Medians are actually not that different and means are within confidence intervals. Perhaps different time periods were used to calculate the averages of the two Sc's?

L292: Above it is mentioned that the footprints are similar but here that they are different? How would the different footprint of the concentration measurement influence the flux?

L297: "Results of IDM fluxes.."

L297-298: Where is this shown?

L300: What are bubbling zones and where are they located? It comes as a surprise here that the chambers are not measured on the footprint of EC. If you measure bubbling zones with lots of ebullition, how is the chamber flux calculated? Don't the bubbles bring sudden bursts of methane, invalidating the normal flux calculation methods?

L300-305: How are large the medians compared to average fluxes?

L332: replace "a month" with "five weeks"

L336: Lower than what?

L344: These are not comparable if not taken from same time periods and same footprints

L365: Excatly, different time periods are compared with each other making the method comparison useless in this form.

L383-386: In the equation there should be FCO2 and FCH4 instead of CO2 and CH4? These are results rather than conclusions. Abbreviation CO2eq is not defined and there are too many significant numbers in the result.

Reference list: Two references are not peer reviewed yet, and there are quite many non-peer reviewed reports included.

References:

Aubinet, M., Grelle, A., Ibrom, A., Rannik, Ü., Moncrieff, J., Foken, T., Kowalski, A. S., Martin, P. H., Berbigier, P., Bernhofer, C., Clement, R., Elbers, J., Granier, A., Grünwald, T., Morgen stern, K., Pilegaard, K., Rebmann, C., Snijders, W., Valentini, R., and Vesala, T.: Estimates of the annual net carbon and water exchange of forests: the EUROFLUX methodology, in: Advances in ecological research, 30, 113–175, 2000.

Mammarella, I., Launiainen, S., Grönholm, T., Keronen, P., Pumpanen, J., Rannik, Ü., and Vesala, T.: Relative humidity effect on the high-frequency attenuation of water vapor flux measured by a closed-path eddy covariance system, J. Atmos. Ocean. Tech., 26, 1856–1866, https://doi.org/10.1175/2009JTECHA1179.1, 2009.

Rantala, P., Taipale, R., Aalto, J., Kajos, M. K., Patokoski, J., Ruuskanen, T. M. & Rinne, J. 2014: Con-tinuous flux measurements of VOCs using PTR-MS — reliability and feasibility of disjunct-eddy-covari-ance, surface-layer-gradient, and surface-layer-profile methods. Boreal Env. Res. 19 (suppl. B): 87–107.

Tedford, E. W., MacIntyre, S., Miller, S. D., and Czikowsky, M. J.: Similarity scaling of turbulence in a temperate lake during fall cooling, J. Geophys. Res.-Oceans, 119, 4689–4713, https://doi.org/10.1002/2014JC010135, 2014.

---

## Author Comment (AC1) · 20 Oct 2020

**Response to comments from Referee #1**

**We thank Referee #1 for the thoughtful comments. The specific questions and suggestions posed, in black, are answered below in blue.**

Detailed comments directed to the authors:
P 4 LL 119-122: You describe that a standard axis rotation was performed within Eddypro. Could you elaborate a bit more on how this rotation was performed? The abrupt terrain change can pose a problem for measurements obtained at an EC station set up at a shoreline. Especially for the wind sectors that might have contributions from land and water surface. Paw et al. (2000) and Finnigan et al. (2003) suggest considering such terrain structures in the rotation procedure of the eddy-covariance data, which can be obtained by a sector wise application of the planar-fit method according to Wilczak et al. (2001).

Response: We used the standard double rotation for the fluxes given in the manuscript (i.e. zeroing the average cross wind and vertical wind components). The slope of the shoreline of the pond was very gentle, and the wind was not expected to experience any significant perturbations near the flux tower. However, to evaluate the reviewer's suggestion, we recalculated the fluxes using a sector wise planar-fit method Wilczak et al. (2001). Four sectors were defined: $286^0$ - $76^0$ (pond sector); $76^0$ - $124^0$ (east shoreline sector), $124^0$ -$259^0$ (the south sector); $259^0$ -$286^0$ (west shoreline sector). The resulting half-hour $CH_4$ EC flux and the original flux were within $0.0 \pm 0.1$ g m$^{-2}$ d$^{-1}$ of each other (mean and standard deviation of the difference). Therefore, as expected, during this campaign at this site the planar fitting method did not significantly change the final $CH_4$ EC flux results.

Section 3.2: Are there any influences of waves to be expected on the calculation of the gradient fluxes?

Response: The pond surface was mostly calm during this study. We observed that the pond surface behaved somewhat differently from natural ponds, since it was partially covered by oil slicks that suppressed wave action. Given the size and shallowness of the pond, waves would have been no more than a few cm in height and therefore insignificant even in relation to the gentle landscape features surrounding the pond.

Section 4.2: Could you please clarify how the shown footprint fits to the flux data set? Particularly I would find it interesting to see a separation of the footprint for the overall data set as well as unstable, stable and neutral conditions. In general an overlay of the entire footprint map over a land use map/aerial photo could provide a more useful inside to interpret the data.

Response: Thank you for the suggestion. We have added the footprint to a revised Fig. 1 and removed Figure S3. As can be seen on Fig.1, the 80% footprint contour lies completely within the liquid water surface of the pond. During this study, 98.6% half-hour periods were associated with unstable stratification when the wind came from the pond. Below, we show footprints under unstable (z/L ≤ -0.0625), neutral (-0.0625 < z/L< 0.0625), and stable (z/L ≥ 0.0625) conditions. We also included more text to describe this in Section 4.2.

[Figure]

You mentioned that one reason for the differences between chamber and EC flux calculations, is the local deployment of the chambers. One further approach to gain more information during a comparison of is to use the Kljun model to calculate the land use contribution for each half hour EC flux. This could help to understand the influence of the mentioned bubbling areas on the flux estimates.

Response: In the revised Figure 1, the locations of the 15 flux chamber measurements were labeled in white circles. They were all well within our 80% footprint, whereas any potential land contributions to the flux are shown by the footprint analysis to likely be insignificant. The bubbling zones on the pond surface were random and cannot simply be distinguished from inactive zones by the surface characteristics from the Google Earth image.

Figure S2: In my opinion it does not add much extra information since there is no clear daily pattern. Maybe a marking which direction represents the pond and land sectors would help.

Response: We agree and have removed this figure.

---

## Author Comment (AC2) · 20 Oct 2020

**Response to comments from Referee #2, Kukka-Maaria Kohonen**

**We thank the referee for this thorough review. Careful consideration of the extensive comments and implementation of many of the suggestions have made this a stronger manuscript. Below, we address each question in turn. Questions and suggestions are in black, and our responses are in blue.**

General comments:
Measurement methods in general require more description:

-Eddy covariance flux calculation description is lacking relevant information. The authors list as correction methods axis rotation, time lag compensation, WPL correction, and storage term correction. Which axis rotation method was used?

Response: More detail has been added to the manuscript (see lines 123-125 in Section 3.1). The standard double rotation (zeroing the average cross and vertical wind components) was applied (cf. Wilczak et al., 2001), and a planar fit method was tested, resulting in insignificant differences. As described in the response to Referee #1, in the test of planar fit method calculation, four sectors were defined: 286º - 76º (pond sector); 76º - 124º (east shoreline sector), 124º -259º (the south sector); 259º -286º (west shoreline sector). The resulting half-hour $CH_4$ EC flux and the original flux were within $0.0 \pm 0.1$ g m$^{-2}$ d$^{-1}$ of each other (mean and standard deviation of the difference). Therefore, as expected, during this campaign at this site the planar fitting method did not significantly change the final $CH_4$ EC flux results.

WPL correction should actually not be applied for this gas analyzer (Picarro G2311-f) as it is already included in the instrument itself.

Response: This was indeed an oversight on our part and we are grateful to the reviewer for pointing this out. We recalculated the fluxes in EddyPro without WPL correction, and found that the new results (half-hour series) are essentially indistinguishable from our original results, with an average decrease of 0.04% lower. Therefore, this correction issue had no significant effect on our results or conclusions.

Spectral corrections are not mentioned in the text. Spectral corrections (especially high frequency spectral correction) are essential in EC flux processing and can affect even the sign (direction) of the flux measurement. Recommended spectral correction methods are introduced in e.g. Aubinet et al., (2000) and Mammarella et al., (2009).

Response: Given the measurement height of 18m, spectral corrections are usually small, which is why we did originally not apply any spectral corrections. Recalculating the fluxes and applying a high frequency correction of low-pass filtering effects according to Moncrieff et al. (1997), we found the new numbers to be very close to the old results. On average, spectrally corrected values were 0.8% higher than uncorrected values. Therefore, this correction did not significantly affect the final pond emission results and conclusions.

Was u* filtering applied? If yes, what was the threshold and how was it determined?

Response: We carefully investigated this issue in our early analysis, and found that there was no evidence of fluxes becoming underestimated or erratic at lower $u_*$ (Figure S4, original Figure S5). For most of the half-hour periods when the wind was from the pond, $u_* > 0.1$ m/s.

How about storage change fluxes, how were they calculated?

Response: Storage fluxes of $CH_4$ were calculated as the second term in equation (2), i.e.

$$F_{storage} = \int_0^z \frac{\partial c}{\partial t} \, \partial z$$

Eddypro assumed that the profile was linear from the measurement point to the ground and calculated the storage flux as a separate term. In this study, the storage flux was added to the calculated EC fluxes in the final EC fluxes. Given that the dynamic stability associated with pond wind directions was in the unstable regime 98% of the time, the storage correction made little difference to the net flux.

Fluxes from different wind directions are presented in this study, but it is not clear weather all these fluxes were processed in similar way. If all wind sectors are covered with different types of roughness elements (such as pond, buildings, trees), the different sectors should be processed (and fluxes calculated) individually. Environmental data required for the flux calculation (air pressure, temperature and humidity) are not described.

Response: Each half-hour flux was calculated independently, and cumulative/average statistics were calculated for the different wind sectors to ensure homogeneous conditions upwind for each sector. Meteorological inputs were described in lines 90-96.

-Gradient flux method has deficiencies.
Eddy diffusivity is calculated from CH4 EC flux, so gradient flux is not totally independent from EC CH4 flux measurements. I understand the eddy diffusivity is not taken directly based on EC measurements, but from a fit of Schmidt number against stability parameter. Even though making this fit makes gradient fluxes not directly dependent from EC, it should still be discussed how the usage of EC measurements in eddy diffusivity calculations affect the comparison between these methods, as it has not been currently discussed at all.
The authors refer to a study by Bolinius et al., 2016 where the eddy diffusivity is calculated from the heat flux measurements of the EC system instead of the gas flux. This is a well established method and I recommend the authors to study it more carefully and implement in their study as well. I suggest the authors at least compare this method to their original gradient flux calculations. Another study worth taking a look at is Rantala et al., (2014), where eddy diffusivity is calculated from the Monin-Obukhov similarity theory. Heat flux is independent from the gas flux, so calculating eddy diffusivity from the heat flux measurements will allow more reliable comparison between EC and gradient CH4 fluxes.

Response: We evaluated the method of using the heat fluxes to establish an eddy diffusivity KT early on, but found these diffusivities to be significantly noisier than those based on momentum. And an obvious problem with using heat fluxes as a baseline is that fluxes are very small at night, and therefore KT becomes very erratic and unusable. Km is also independent of the gas fluxes and has the advantages of being relatively well-behaved and continuous. There are similarities between our approach and that of Rantala et al (2014), but our approach does not rely on Monin-Obukhov Similarity Theory since we use the directly measured momentum flux, and the stability corrections that are explicit in the M-O approach are incorporated into our formulation of the Schmidt number.
We are well aware that our gradient fluxes are not truly independent of the eddy covariance fluxes. However, the fact that the Schmidt number we calculate agrees with previously published constant serves as an independent verification of the gradient flux approach. Even using a constant Schmidt number and stability corrections from literature, i.e. not using the measured EC methane fluxes at all, would have produced very comparable gradient flux numbers.

- Chamber measurements are currently not described at all but a proper method description is needed (what kind of chamber design was used, dimensions, how long enclosure time was after reaching

equilibrium with carrier gas flow and inside air, how was the air flow implemented, how was the flux calculated, what kind of data selection methods were used etc.).

Response: The chamber measurements, which were performed by a third party independent of our project, followed the US EPA Standardized point measurement technique (adapted from Kienbusch, M., Measurement of Gaseous Emissions rates from Land Surface Using an Emission Isolation Flux Chamber, User Guide, EPA Users Guide, Contract No. 68-02-03389-WA18 (EPA/600/8-86/008), 1986. Regulations regarding chamber measurements in Alberta are given in https://open.alberta.ca/publications/9781460145814. The key steps are reproduced here:
1.      Once the flux chamber (~0.1m$^2$ surface coverage area) is deployed on the target surface of interest, the valve of nitrogen cylinder was opened to begin purging the flux chamber with 99.9995 percent pure nitrogen gas. The flow rate of the nitrogen sweep gas was adjusted to a certain flow rate using the rotameter and this rate was maintained throughout the sample duration. The exhaust gas sample/purge rate did not exceed 2.5 L/min. This prevented ambient air entraining into the chamber and maintained a minimum exhaust rate 2.5 L/min out of the pressure equalization port. The GHG analyzer has an internal pump that operates at 0.5 L/min. The start time of the purge and the initial concentrations of CH4 were recorded on the field data sheet.
2.      For the first 45 minutes, concentration readings were noted on the field data sheet in 15-minute intervals. Approximately five times the flux chamber volume was be purged in the first 45 minutes of sampling. Generally, the $CO_2$ and $CH_4$ concentrations reached steady-state 30 to 45 minutes into purging as indicated by a plateau in the real time data curve.
3.      After 45 minutes, concentration readings were recorded every 10 minutes for a minimum of 75 minutes of total sampling duration. A minimum of 30 minutes of steady-state concentration data had to be obtained for the sample to be valid. The recorded concentrations and times on the field data sheet act as a back up to the GHG analyzer data files.

Comparison of fluxes is highly misleading and fundamentally flawed. The authors have included in the flux averages all data available, which are then compared with each other. What the authors should do instead is to select only those time periods/data points when all the measured fluxes are available, and then calculate averages that are comparable. If this is not done, it easily happens so that one of the methods is measuring e.g. more fluxes from one wind direction or time of the day than the other, which is causing a clear bias in the comparison.

Response: We have added text to clarify that the comparisons we show for the EC fluxes, gradient fluxes and inverse dispersion model fluxes are of course based on the set of simultaneous half-hour measurements over the 5-week study, when data was available to calculate all three. This does not apply to the comparison with the chamber fluxes, since the flux chamber measurements, conducted by a third party, happened to be performed when the wind was from the south and the micrometeorological methods (located on the south shore) could not observe the pond. For this comparison, the assumption is made that emissions from the pond are relatively time invariant during the period that was missed by the micrometeorological fluxes, as supported by the time series of fluxes for wind directions from the pond during the study period. This is in fact a common assumption made in many applications of flux chamber work, due to the snapshot nature of such measurements, and represents a significant limitation of flux chambers that we highlighted in lines 44-48 and in section 4.6 and 4.7. These well-known limitations were one of the reasons for exploring alternative methods for quantifying fluxes from such sources of fugitive emissions.

Conclusion section is currently an additional discussion section that should have been implemented in the section "Results and discussion" already. Proper conclusions – with no new information given but rather a summary with a perspective to future studies – is totally missing and should be included.

Response: A conclusion section has been added, and a comparison to previous results was inserted into Section 4.7.

Specific comments:

Table 1: This comparison does not make sense if the fluxes are not averaged from simultaneous measurements. You should only include the datapoints in averaging when you have a datapoint from all the methods. What does it mean that fluxes are "relatively steady"? The uncertainty estimation in footnote c is unclear.

Response: As explained above, the comparison of EC flux, gradient flux, and IDM flux is of course based on a set of simultaneous half-hour data points, when data was available to calculate all three, with wind directions from the pond. The exception are the flux chamber results, for which an assumption of time invariant fluxes during the concurrent micrometeorological flux time series is required.

The uncertainty estimation is based on a conservative, integrated approach encompassing all errors. In the real data time series, periods were identified when the flux did not fluctuate much, i.e. represented steady state conditions. In this case, we found in EC flux time series five periods of at least five half-hours with standard deviation not greater than 0.89. Then, the average of the standard deviations from these five periods was used as the uncertainty of EC flux in this study. For gradient flux and IDM flux, we used the same five periods, and calculated the average of the standard deviations from the five periods. This approach provides an upper limit or conservative estimate of the overall uncertainty in the final flux.

Table 2: Not clear what are the time periods for these flux estimates, should they even be comparable? Annual averages are different from summertime measurements. It would be interesting to see a comparison to natural waters or reservoirs as well, to see the high magnitude of the methane emissions.

Response: We are comparing all data available publically for this particular pond. The Small et al. (2015) data were from measurements in 2010 or 2011. Stantec report (2016) data were from measurements in 2013 and 2014. Baray et al. (2018) results were from the aircraft campaign in 2013, which was discussed in Section 4.7. These data were compared to provide context of results from this study. The comparison of annual averages vs. summertime measurements was discussed in Section 4.7, in the first paragraph. Natural lakes and indeed even wetlands emit at rates well below what we observed on this industrial pond, typically on the order of 0.005-0.05 g m$^{-2}$ d$^{-1}$ (Sanches et al., 2019).

Figure 1: It would be very helpful for the reader to include in the map the EC footprint lines and/or lines for approved wind directions. It is not very clear from the closeup image where exactly are the pond edges. Maybe this could be highlighted somehow? Add chamber measurement locations to this map.

Response: Yes, the EC footprint lines and chamber measurements have been labeled in the revised Figure 1. A sentence has been added to the caption explaining the different shadings of surface cover.

Figure 2: What is the correlation coefficient of the linear fit? How does it change if you use the original datapoints instead of binned averages for fitting? It does seem that data are very scattered with higher Km and Kc, how does this affect the fitting? What do the boxplots represent (what are the box limits, whiskers, center line etc)?

Response: The correlation coefficient for the linear fit was r$^2$=0.93. The original data points were indeed quite scattered, but produce the same slope. The figure caption has been revised to label box, lines and whiskers.

Figure 3: "Best fit" - determined by what criterion? The bins are not of equal size and I believe this is also affecting the fit. What do the boxplots represent (what are the box limits, whiskers, center line etc)?

Response: Bins are actually of equal size with bin width = 1, except for the last bin on the right for z/L > -0.34. At z/L=0.34, the exponential is equal to the Sc=0.923 found from the linear fit in Fig. 2; therefore, this was chosen as the point to switch from the exponential to the constant part of the function.

Figure 4: Fig 4b is not discussed anywhere and is a bit pointless without water temperature. In 4a, u* is missing interquartile ranges and 10% and 90% percentiles. In 4e sensible heat flux is missing quartiles off-pond and 10% and 90% percentiles on pond. Mark in the diurnal plots the times of sunset and sunrise to help the reader.

Response: All the information required is shown; from the temperatures in panel (b) and the temperature difference shown in (c), the absolute pond surface temperature can be inferred if desired. Panel (b) was in Figure 4 to show the typical diurnal cycle of ambient temperature. We have now cited panel(b) in Section 4.1.
The figure has been revised to include sunrise, sunset and off-pond heat flux quartiles.

Figure 5: Scale seems quite arbitrary, how was it defined? Directions are missing, where is north?

Response: The scale was set to include roughly equal numbers of data points in each range. The figure caption has been revised to label north.

Figure 6: You should add a, b and c to subplots. Colors of EC and gradient fluxes are too similar in the printed version and in the lowest panel red and green are used which is not color-blind friendly. You can check colorblind and printer friendly color choises e.g. from here: https://colorbrewer2.org

Response: Fixed.

Figure 7: Shade the pond area also here, similar to Fig. S6. What do the boxplots represent (what are the box limits, whiskers, center line etc)?

Response: Shade has been included here. Lower and upper bounds of the box plot are $25^{th}$ and $75^{th}$ percentile; the line in the box marks the median and the black square labels the mean; the whiskers label the $10^{th}$ and $90^{th}$ percentile.

Figure 8: What is the offset of the fit? It does not seem to be crossing y=0 at x=0 in neither of the plots.

Response: They both indeed cross (0,0).

Figure S2: What do the confidence intervals represent?

Response: The blue shade is $25^{th}$ percentile to $75^{th}$ percentile of the wind direction in degrees. This figure shows that the diurnal variation of wind direction is weak. However, this point can be made in the text without a supporting figure, as suggested by the other reviewer, and we decided to remove this plot.

Figure S3: "...countours of the EC footprint area". It would be very helpful for the reader to get S3 b on top of a map, to see where the contours are crossing pond edges.

Response: We agree. The footprint contour is now superimposed onto the pond map in the revised Figure1. FigureS3 has been removed.

Figure S4 (now S2): It is not mentioned here which EC flux this is. Methane? Mention in each subplot which wind direction it is representing (in legend/title/xlabel/ylabel) to help the reader. What do the boxplots represent (what are the box limits, whiskers, center line etc)? Mention in the caption what is in each wind sector (pond, buildings, trees, etc).

Response: Yes, we mean methane EC fluxes. All these suggestions are accepted and Figure S2 has been updated. Lower and upper bounds of the box plot are $25^{th}$ and $75^{th}$ percentile; the line in the box marks the median and the black square labels the mean; the whiskers label the $10^{th}$ and $90^{th}$ percentile.

Figure S5 (now S3): Mention in the ylabel that this is methane flux. Mention in the caption what is the r2 representing (least squares linear fit?).

Response: Caption has been revised to note $r^2$.

Figure S6 (now S4): What do the boxplots represent (what are the box limits, whiskers, center line etc)?

Response: Caption has been revised.

Figure S7 (now S5): What do the boxplots represent (what are the box limits, whiskers, center line etc)?

Response: Caption has been revised.

Table S1: Are the fluxes compared here from exact same time periods? Same comment as for Table 1 about the uncertainty estimate and "relatively steady".

Response: Yes, the $CH_4$ gradient flux with variable Sc and constant Sc use exactly the same vertical mole fraction gradient data over exactly the same set of simultaneous data. An explanation of the uncertainty estimate was given in our response to comments on Table 1 above.

L10: "develop" is a little bit misleading here since the authors don't really develop any new method, rather compare already existing ones.

Response: While all three micrometeorological methods are of course well established, we are not aware of any previous instances of our approach of calculating the gradient fluxes through the use of a momentum flux diffusivity adjusted with a stability-dependent Schmidt number. Therefore we would like to keep the current term.

L11-12: Mention briefly which are these three flux methods in one sentence.

Response: Done.

L15: inverse dispersion model comes here from out of the blue. Describe it briefly before writing about the results.

Response: Done. The Inverse dispersion model is now introduced in line 12. A detailed explanation of the method is given in section 3.3.

L18-19: This sentence is a bit misleading. In one perspective it is quite obvious that a larger footprint represents a larger area. On the other hand if the EC tower is placed so that it is measuring only e.g. shallow area while actually the pond is deeper from a much larger area, then would EC be representative of the whole pond emissions? Then on the other hand nobody can know what is the real flux. It might as well be closer to the chamber flux than EC.

Response: There are several reasons that point to the eddy covariance fluxes being the more accurate estimate of the true fluxes. We have shown that fluxes were consistent for various wind directions across the pond, over a month of measurements, and they represent a large fraction of the pond surface. This is in contrast to the chamber measurements which cover a total of a few $m^2$ for instantaneous snapshot measurements, limited to regions of the pond accessible by boat. Implications have been discussed in the manuscript section 4.6 and section 5. Also of concern is the large interannual variability in flux chamber results, with 5.3 g $m^{-2}$ $d^{-1}$ in 2016, 2.8 g $m^{-2}$ $d^{-1}$ in 2017 and 11.1 g $m^{-2}$ $d^{-1}$ in 2018, despite similar operational conditions.

L21: Abbreviation AOSR is not used anywhere in the text

Response: Deleted.

L23: "Oil Sands" or "oil sands"? Throughout the manuscript.

Response: Made "oil sands" consistently throughout.

L48: "eddy covariance (EC)" and then use EC after this throughout the manuscript instead of eddy covariance

Response: Fixed.

L49: "area sources" or "source areas"?

Response: Area sources

L53: So only emissions can be measured with this method, not uptake?

Response: Uptake can of course be measured too, and would manifest itself as a negative flux. The two cited studies were emission flux studies.

L56: What is meant by "relatively well-defined spatially"? If the fluxes are well-defined, why do you measure them?

Response: We meant that the source area was relatively well-defined spatially, not that the fluxes were known. The ponds cover a well-known spatial domain, in a remote region far from urban activities and other sources.

L59: "Field study" is not a very descriptive title. Maybe "Site and measurement description"?

Response: Agreed & implemented.

L62: Trees are not part of natural landscapes? What is? How far were the other facilities? In the catchment area or further away? How large is the cathment area?

Response: This artificial pond is elevated above the surrounding landscape and has minimal catchment (~ 100m around the shoreline). The influx of industrial processed water vastly dominates the water budget of the pond. We added distances to the main facilities nearby to the text in lines 62-65.

L65: What is meant by "mobile tower"? How high were the measurements above water (which is more relevant than ground in the case of pond fluxes)?

Response: The tower mounted on a truck bed and can be easily towed from place to place for temporary installations. The base of the tower was less than 30cm above the water surface.

L69: Is this the diameter or radius? Inner or outer diameter?

Response: Outer diameter. This is now noted in line 73.

L71-72: I am not sure it can be said that turbulent flow is ensured. Reynolds number is _1300 according to my calculations, so it is possible that the flow is turbulent, but I wouldn't call it "ensured".

Response: Teflon tubing is generally labelled by outside diameter. Calculating the Reynolds number with the inside diameter of (3/8" minus wall thickness 1/16") D = 0.635 cm, with a flowrate of Q =117 cm$^3$ s$^{-1}$ and a kinematic viscosity of $\nu$ = 0.148 cm$^2$ s$^{-1}$ gives Re = (Q D)/($\nu$ A) = 4500, well within the turbulent regime.

L72-74: All kinds of measurements are presented that are not used in the analysis or shown anywhere in the manuscript. I suggest to leave out the description of those gas measurements not used in this particular study. Why is a 40 m long tubing required for 18 m height measurements? This will cause quite long lag time for EC. What are the three and four levels mentioned here?

Response: 40m tubing was used for all gradient levels including the 18m height measurement, to avoid systematic differences due to tube lengths. Three levels was a typo; the G2204 sampled from four levels, 8m, 18m, 32m on the tower plus 4m on the roof of the trailer. The length of the EC 3/8" OD line was 30m; this has now been added to the manuscript. We have removed the description of the G2401-M analyzer since its data was not used in this work.

L74: There must be some flush time of the tubings and analyzer between the different height measurements. How long is the flush time? One level cannot be measured 2.5 min during 10 min period if you take into account the flush time.

Response: Air was drawn through all 4 tubes continuously, and the only part of the flow system requiring flushing was the last 4m of tubing. To allow for the flow and pressure to equilibrate after each level switch, the first 30 s of the 150 s period at a given level were eliminated from the averaging process.

L76-77: Was there any drift of the instruments between calibrations? Did they compare well with each other?

Response: The calibration coefficient (slope) for CH$_4$ changed by 0.12% from before to after the study, and the offset by less than 0.002 ppm.

L80-82: This is well known EC theory and does not need to be explained.

Response: Since this information is fundamental to this paper and not all readers of this journal may be familiar with it, we chose to retain this.

L88-89: Was the infrared sensor calibrated somehow?

Response: No.

L92: How were the suitable wind directions determined?

Response: They were based on the map.

L104: EC also has its limitations, "benchmark" seems a bit exaggerated

Response: We have changed "benchmark" to "reference".

L106: Response time and sampling frequency are not the same. Response time should be given in seconds, sampling frequency in hertz. EC measurements require both fast response times and high sampling frequency.

Response: Fixed.

L107: "CO2 and CH4 fluxes"

Response: Fixed.

L108-109: Reformulate the sentence. EC does not calculate anything, and in this case you are talking about gas fluxes explicitly (not e.g. heat flux since you mention mole fraction)

Response: "method" is inserted after "eddy covariance".

L113-114: Repetition from above

Response: Modified.

L115: "storage change flux". Out of curiosity, how large was the storage change flux? Often in lake studies they have been neglected but might be important as well.

Response: $CH_4$ storage fluxes was small. When the wind was from the pond, the storage flux was -2% to 3% (interquartile range) of the first term in equation (2).

L118-122: More description is needed on the processing methods used. How long was the lag time on average?

Response: More description has been inserted.
Covariance maximization method was used in time lag compensation. This method maximizes the covariance to variables (Fan et al. 1990), within a window of plausible time lags automatically calculated by EddyPro. The lag time on average was 12 second. (Fan et al., 1990).

L122: What do the different flags mean (what are the criterion)?

Response: As described in Mauder et al. (2016) and Mauder and Foken (2004), the quality test calculates the ratio of the standard deviation of CH4 flux to CH4 flux. Then, this ratio (or relative standard deviation) is compared to modelled results (as described in Mauder and Foken (2004)) to get a relative

difference. The flags are determined based on this relative difference. In this study, we used the widely used overall flag system also described in Mauder and Foken (2004): flag = 0 when this relative difference < 30%, flag = 1 when 30% < this relative difference < 100%; and flag = 2, when this relative difference > 100%.

L123: "Gradient flux method"

Response: Fixed.

L130: units?

Response: Unit of Kc are $m^2s^{-1}$

L135: How do you define the gradient method footprint?

Response: This was mentioned lines 136-137 and explained in detail in the footprint section (4.2).

L168: Shifting winds are also a problem for EC measurements!

Response: That is correct. Since shifting winds are a problem for both methods, our way of excluding fluxes when the signs of EC flux and gradient fluxes were opposite at least partially excluded such situations from the comparison.

L177-180: What are the units of these variables?

Response: In this study for $CH_4$, unit of C and $C_b$ is ppm, unit of Q is $g\ m^{-2}day^{-1}$. However, the formulation is valid for any consistent system of units desired.

L181: Why are L and u* used as inputs, since u* is already used in calculating L?

Response: To quantify stability, $u_*$ by itself is insufficient, and the heat flux is also required, as incorporated in the Obukhov length L. Since L was already defined in the previous section, stating the inputs in this manner seemed the least confusing option to us.

L185: How close is "right beside"?

Response: Fixed. The new sentence " $CH_4$ mole fraction input was taken from the OP-FTIR measurement which was located 10m to the east of the flux tower."

L188-195: More description needed

Response: Yes, we have inserted more details. The full details are available through the Alberta online public report cited, and are not the focus of this paper.

L196-199: Not understood where this is used (which methods) and why. More description please.

Response: The standard method for flux chamber operations for compliance monitoring in Alberta was used. The details can be found in the online report cited in Section 3.4. Please also see the response above for the chamber measurements.

L202: "wind coming from" or "wind that came from"

Response: Changed to "coming from"

L214: Sunsets and sunrises are not directly seen from Fig 4d.

Response: Fixed. The revised Figure 4 now has yellow shades marking the range of sunrise and sunset times for the study.

L215-216: The same wind direction/ source area applies to gas fluxes. Why different wind direction analysis is applied to sensible heat fluxes and not the others?

Response: This section was talking about meteorological parameters. All fluxes, including sensible heat fluxes and gas fluxes, were analyzed in the same manner. Results in Table1 are based only on data associated with wind directions from the pond.

L229: You can only see this from Fig 4c. What is meant by "species"? Methane?

Response: This applies to all species emitted from the pond, not just $CH_4$.

L225-230: This is done for wind direction filtered data I assume? Should be clarified since all wind directions are analyzed in some way or another in this manuscript.

Response: This has been done for the all half-hour periods. The footprint polar plot shows the footprint under unstable conditions. This is noted more clearly now in the revised Figure 1 caption.

L230: This is not seen from Fig S3 (b), since you cannot see pond edges in the figure.

Response: Fixed, in the revised Figure 1.

L231: "gradient flux"

Response: Fixed.

L234: delete "however"

Response: Fixed.

L235-240: But there is much more data from off-pond direction than from pond direction. How does influence the analysis?

Response: That was the reality dictated by uncontrollable constraints such as site accessibility and weather, and one reason for scheduling a 5-week long campaign, to ensure a statistically significant number of days for most wind directions. There were 280 half-hour periods when the wind was from the pond and 98.6% of them were under unstable conditions, i.e. our footprint results have reasonable statistics.

L245-253: It is not clear why fluxes off-pond are reported since this is a study concentrated on pond emissions. Are these sectors processed in flux calculation individually or not?

Response: All the sectors were processed in the same way in EddyPro calculation. Off-pond fluxes were reported here to provide a measure both of the methodological noise in the signal, as well as the background (non-pond) flux magnitudes. Off-pond fluxes are either close to zero, or had a slight increase during the middle of day. The pond fluxes had no significant diurnal pattern.

L253-258: Are these now results from pond direction? Wind and turbulence are still driving the turbulent/diffusive transport of gases from pond to the atmosphere (e.g. Tedford et al., 2014).

Response: Yes, fluxes from pond direction. The pond flux diurnal cycle was shown in Figure S4(a). In the main manuscript we state "The lack of a diurnal variation of $CH_4$ EC flux observed when the wind was from the pond in this study was similar to the lack of a diurnal variation of $CH_4$ EC flux at another tailings pond reported by Zhang et al. (2019)."

L263-265: How do medians correlate? Take into account my earlier comments about representativeness as well.

Response: According to the numbers in Table 1, the median of $CH_4$ gradient flux is 20% lower than median EC flux. Again, the EC fluxes and gradient fluxes cover exactly the same periods and were calculated using the exact same measurements.

L270-272: This is quite far taken conclusion. Based on the results here you can only say that EC fluxes were used to calculate Kc, which of course then correlates well with EC.

Response: The idea behind this statement is that the $K_c$ calculated in this way lets us calculate gradient fluxes for any species emanating from the pond that obeys the same physics (turbulent transport) and chemistry (inertness) as $CH_4$. Most gradient methods (modified Bowen, aerodynamic etc.) depend on some input of an EC flux, for example through $u_*$ or the sensible heat, so there will always be some measure of autocorrelation. In our case, the link to the EC fluxes of methane are strictly through the parameterization of the Schmidt number, so a perfect correlation is not a given.

L272-274: And what were the outcomes of these studies? How do they compare to this study?

Response: These sentences were to show to compare our findings to the surprisingly few previous studies which compared EC flux and gradient flux of $CH_4$: "Zhao et al. (2019) compared $CH_4$ fluxes from an MBR method as well as from an aerodynamic flux model to EC fluxes for two small fish ponds, and showed that the MBR fluxes were well correlated with EC fluxes, with a mean 27% greater than the EC mean flux." Such studies are rare, so we feel our contribution represents a useful addition to the body of studies investigating EC, eddy diffusivity, and gradient fluxes.

L278-281: Medians are actually not that different and means are within confidence intervals. Perhaps different time periods were used to calculate the averages of the two Sc's?

Response: The average of two Sc methods used exactly the same set of half-hour periods, which are the entire period of this campaign. With stability z/L corrected Sc, the mean gradient flux is closer to mean EC flux, compared to using the constant Sc.

L292: Above it is mentioned that the footprints are similar but here that they are different? How would the different footprint of the concentration measurement influence the flux?

Response: For infinite homogeneous upwind fetches, the methods have the same footprint, given correctly placed gradient levels. In real-life situations, there can be differences, since the upper gradient

point has a larger concentration footprint than the lower one, and therefore may see sources farther upwind.

L297: "Results of IDM fluxes.."

Response: Fixed.

L297-298: Where is this shown?

Response: It is shown in another AMTD manuscript: https://amt.copernicus.org/preprints/amt-2020-257

L300: What are bubbling zones and where are they located? It comes as a surprise here that the chambers are not measured on the footprint of EC. If you measure bubbling zones with lots of ebullition, how is the chamber flux calculated? Don't the bubbles bring sudden bursts of methane, invalidating the normal flux calculation methods?

Response: The locations of the 15 flux chamber measurements are marked in the revised Figure 1. As can be seen, most of them fall within the EC footprint. It is possible that the sudden bursts of $CH_4$ could invalidate the flux chamber calculation and lead to an underestimation of flux, as discussed in Zhang et al. (2019). We wrote this in line 334-337. Integrated over the footprint of micrometeorological flux measurements, the intermittent nature of ebullition will have a minimal effect.

L300-305: How are large the medians compared to average fluxes?

Response: The median of the 15 measurements is 2.3 g m$^{-2}$day$^{-1}$, and the mean is 2.8 g m$^{-2}$day$^{-1}$. In addition, the 15 measurement fluxes were scattered indicating "the pond was highly heterogeneous in terms of $CH_4$ emissions".

L332: replace "a month" with "five weeks"

Response: Fixed.

L336: Lower than what?

Response: Lower than results from three micrometeorological methods. Fixed.

L344: These are not comparable if not taken from same time periods and same footprints

Response: As explained in our response in the General Comments section above, the comparison between the chamber and micrometeorological flux measurements requires the assumption that emission rates from the pond did not change much in the few days before, during and after the chamber sampling. This assumption is supported by our CH4 flux time series. We included this comparison to put results of this study into context of historical data and current operational monitoring and reporting methods, and to shine the light on future monitoring needs, such as seasonal variability of tailings pond emissions. We acknowledge in the text that "This reflects a general complication when comparing the five weeks emission results in this study to annual emissions reported in the past."

L365: Excatly, different time periods are compared with each other making the method comparison useless in this form.

Response: Please see the response to the previous comment.

L383-386: In the equation there should be FCO2 and FCH4 instead of CO2 and CH4? These are results rather than conclusions. Abbreviation CO2eq is not defined and there are too many significant numbers in the result.

Response: Fixed.

Reference list: Two references are not peer reviewed yet, and there are quite many non-peer reviewed reports included.

Response: It is an unfortunate reality that there is very little information in the peer-reviewed literature on industrial fugitive emissions to the atmosphere in the Alberta Oil Sands, which is one reason for the importance of this current manuscript. This lack of published information makes it difficult to avoid referring to grey literature. We have removed the two references to the yet unpublished manuscripts by Moussa et al., and updated the information on the You et al. manuscript on FTIR measurements in AMTD.

**References**

Fan, S. M., Wofsy, S. C., Bakwin, P. S., Jacob, D. J. and Fitzjarrald, D. R.: Atmosphere-biosphere exchange of $CO_2$ and $O_3$ in the Central Amazon Forest. Journal of Geophysical Research, 95: 16851-16864, 1990.

Mauder, M., and Foken, T.: Documentation and instruction manual of the eddy covariance software package TK2, https://core.ac.uk/download/pdf/33806389.pdf, 2004.

Mauder, M., Liebethan, C., Göckede, M., Leps, J-P, Beyrich, F., and Foken, T.: Processing and quality control of flux data during LITFASS-2003, Bound-Lay. Meteorol., 121, 67-88, 2006.

Moncrieff, J. B., Massheder, J. M., de Bruin, H., Ellbers, J., Friborg, T., Heusinkveld, B., Kabat, P., Scott, S., Soegaard, H., and Verhoef, A.: A system to measure surface fluxes of momentum, sensible heat, water vapor and carbon dioxide, Journal of Hydrology, 188-189, page 589-611, 1997.

Sanches, L. F., Guenet, B., Marinho, C. C., Barros, N., and de Assis Esteves, F.: Global regulation of methane emission from natural lakes. Scientific Reports, 9, (1), 255, 2019.

---

## Author Response (AR2)

**Response to comments on the revised manuscript, December 2020**

The authors have made some improvements to the previous version of the manuscript and answered satisfactorily to some of my previous comments, but the bigger comments from last comment round were not taken into account properly. There are still four major issues (in addition to few smaller ones) that require addressing before publication in AMT.

Response: We are very grateful for the reviewer's persistence, which prompted us to re-examine our analysis and to discover two issues that we have now corrected. The first was incorrect accounting for the delay time of the gas concentration signal relative to the sonic anemometer, the second was incorrect application of spectral corrections. The combined corrections resulted in an increase of the fluxes by 25%. Details are given below. Since these adjustments in the eddy covariance fluxes were not constant in time and therefore do not translate into a simple scaling of the gradient fluxes, changes in most of the figures will be noticed, as well as in the comparisons between methods.

We also thank the reviewer for insisting on additional statistical tests to determine the significance of the comparisons. These tests made it clear that the gradient flux method is not very strong at predicting fluxes from one half-hour to the next, and that averages of fluxes binned with by wind direction sectors or by hour of day)is required to produce statistically significant agreement. We have moderated our statements in the abstract and conclusion accordingly.

1) The manuscript and its analysis would really benefit from statitistical tests used to check wether the fluxes measured by different techniques really differ or not. The amount of chamber measurements is probably not enough for this purpose, but fluxes from EC, IDM and gradient flux methods could certainly be used for such tests. You should also test if the nighttime and daytime CH4 fluxes really are statistically the same or not.

Response: we thank the reviewer for this suggestion. Two-sample paired t-tests were applied to the half-hour fluxes or average fluxes binned with wind direction sectors from EC, IDM and gradient methods, when the wind was from the pond.

Table 1: t-test of fluxes from three methods when the wind was from the pond

| Variable 1 | Variable 2 | p-value (half-hourly) | p-value (binned by wind direction sectors) |
|---|---|---|---|
| EC flux | Gradient flux | 0.003 | 0.30 |
| EC flux | IDM flux | $< 10^{-15}$ | 0.08 |
| Gradient flux | IDM flux | $< 10^{-4}$ | 0.33 |

To test if there was a statistically significant difference between daytime and nighttime fluxes, half-hour fluxes were separated by day and night, defining day as 6:00 to 21:00, and night as 22:00 to 5:00 MDT (local time). With t-test results on average fluxes binned with wind direction sectors, day average fluxes and night average fluxes are statistically the same.

Table 2: t-test of fluxes from three methods when the wind was from the pond, fluxes were binned by wind direction sectors and weighted by pond sector area.

| | Flux during the day (mean) | Flux during the night (mean) | p-value (half-hourly) | p-value (binned by wind direction) |
|---|---|---|---|---|

| EC flux | 7.4 | 8.2 | 0.16 | 0.54 |
|---|---|---|---|---|
| Gradient flux | 7.2 | 7.6 | 0.04 | 0.94 |
| IDM flux | 4.9 | 7.6 | $< 10^{-5}$ | 0.04 |

We have modified the text to reflect that (1) EC, gradient, and IDM mean fluxes are statistically different on the half-hour scale, and statistically agree better after averaging with wind direction sectors, when the wind was from the pond; (2) fluxes during the day and at night are statistically not different, although there  was difference in IDM fluxes between day and night time

In Section 4.3, we modified the first sentence in the second paragraph: There was no statistically significant diurnal pattern of the $CH_4$ EC flux when the wind came from the pond direction ($WD \geq 286°$, or $WD \leq 76°$) (relative standard deviation is 15%, p=0.54) (Fig. S4 (a)).

In Section 4.4, we modified the text to read: "Due to significant scatter, the half-hour gradient fluxes were statistically different from the EC fluxes when the wind was from the pond direction (p-value=0.003). They were moderately correlated (slope=0.80, r=0.32, Fig. S7(a)). To obtain some comparability, it is therefore necessary to average blocks of data into appropriate bins. A t-test of the gradient and eddy average fluxes binned by wind direction (22.5° blocks) yielded a p-value of 0.30, and hourly diurnal averaged fluxes agreed with a p-value of 0.09. The pond area weighted mean gradient flux was 8% less than EC flux, and the median was 18% less than EC flux (Table 1)."

In Section 4.5, we modified the text to "IDM and EC flux showed reasonable correlation (r=0.62) with a slope of 0.69 (Fig. S7(b)), although the averaged half-hour IDM fluxes are significantly different from EC fluxes ($p<10^{-4}$). Binning into 16 wind direction sectors similar to described in Section 4.4 yielded agreement at the p=0.08 level. The pond area-weighted mean IDM flux was 30% smaller than EC flux, and the pond area-weighted median IDM flux was also 30% smaller than the EC median flux. The IDM flux showed weak diurnal variations when the wind came from the pond directions (Fig. S8), with smaller fluxes during the day, compared to fluxes at night (p = 0.04), inconsistent with EC and gradient fluxes. As stated in Section 3.3, half-hour periods when $u_*<0.15$ m/s were excluded in IDM calculation (Flesch at al. 2004). This filtering excluded more nighttime fluxes than daytime fluxes, which caused more limited data in IDM nighttime fluxes and biased the t-test."

In the Conclusion, we modified the text: "The gradient and inverse dispersion methods agreed moderately with EC results (18% and 30% lower, respectively)".

2) About spectral corrections of the flux measurements. The authors responded to the request of spectral corrections that the high frequency spectral correction is not important at a measurement height of 18m and changed the average flux by 0.8%. While it is true that the importance of smaller eddies decreases at higher measurement heights, they still cannot be neglected at 18m (not high enough and smaller eddies probably still exist). In addition, the importance of low frequency spectral correction increases with higher measurements, and should be accounted for in the analysis. Why are the spectral corrections still not included in the measurement description? What kind of spectral correction method(s) did you use? You should check the power spectra of your measurements to check if high- and/or low frequency spectral corrections are needed. I am not conviced by only comparing the average flux, without low

frequency correction. If you still are omitting the spectral corrections, you should justify it in the text very clearly and include the power spectrum and cospectrum that proves it.

Response: We are thankful for the reviewer's persistence, since a more careful review of the application of previous spectral corrections uncovered some problems with our usage of EddyPro, which we have now addressed. Cospectral analysis indicated that rather than the minimal effect of < 1% we previously stated, spectral losses actually accounted for an average of 10%. A plot has been added to the supplemental information (S1) to show the average normalized cospectral densities for $CH_4$, $CO_2$ and sensible heat for those periods when the wind was from the pond and the data quality flag for the $CH_4$ flux, according to Mauder and Foken (2004), was either 0 (best quality) or 1 (good quality). Rather than plotting the cospectrum as a function of a normalized frequency (nz/u), we chose to use the natural frequency since high frequency losses will more likely be directly tied to natural frequency. Also, we chose a linear y-axis to facilitate the estimation of losses, which in this format are proportional to the missing area under the curve. High frequency corrections were applied using the approach outlined in Horst (1997). Low frequency spectral corrections according to Moncrieff et al. (2004) were applied, but did not significantly affect the $CH_4$ flux, as can be deduced from the relatively clean cospectral shape and similarity to the sensible heat cospectrum below 0.01 Hz (Fig. S1). It is also clear that the same cannot be said for $CO_2$, which is more frequently affected by signals varying over time scales greater than ~ 10 minutes and less likely to be due to turbulent flux from the (pond) footprint. This is also a good confirmation that the pond is the dominant source of essentially the complete $CH_4$ cospectrum, and that background fluctuations or sources outside the footprint are unlikely to significantly affect the $CH_4$ fluxes.

The power spectra, shown here, also indicate that a high frequency drop-off for the gas measurements in the inertial subrange faster than that of temperature, starting at normalized frequencies as low as 0.3, corresponding on average to a frequency of 0.06 Hz:

[Figure]

To provide a summary of the spectral analysis and corrections performed, we added the following text to section 3.1, last paragraph:

"Covariance spectra were examined for signal losses at higher frequencies (smaller eddies) during transit of the sampled air through the sample line, finite sample cell volume, and instrument response (Fig. S1), accounting for a loss of typically 10% of covariance signal, compared to the sensible heat cospectrum that does not suffer from equivalent losses. Spectral corrections following Horst (1997) were applied to correct for these losses. Low frequency losses at the low frequency end of the spectral peak due to the finite averaging time were applied according to Moncrieff et al. (2004). The EC flux quality flag was categorized into 3 classes: 0 (best quality), 1 (good quality), and 2 (poor quality) (Mauder et al., 2006; Mauder and Foken, 2004). Only EC fluxes with flag 0 or 1 were included in further analysis."

3) The fact that the gradient flux measurements are relying heavily on EC CH4 flux measurements is still not discussed with the flux comparison results. I well understand that the gradient flux was calculated based on a fit to CH4 EC fluxes and not directly to the fluxes, but there is still a strong link. What would make it a bit more reliable is to test the methods I recommended in the previous comments or make the same fit you have now done but using CO2 fluxes instead of CH4, and then calculate the gradient fluxes in similar fashion. In any case, it should be discussed in the results how your calculation method affects the comparison!

Response: The $CO_2$ flux, as seen in the average cospectrum in Fig. S1, was not nearly as well-defined at the $CH_4$ flux in this location, due to significant diurnal background variability as well as the presence of various $CO_2$ sources and sinks in the surrounding area, whereas for $CH_4$ there was only one dominant source (as supported by the wind direction dependency as well as the cospectra). This is what made this location a good place to relate gradients to gradient fluxes, using $CH_4$. Also, the $CO_2$ fluxes out of the pond were, relatively speaking, significantly weaker than the $CH_4$ fluxes, as can be illustrated with this figure showing about 3 days of data:

[Figure]

The CO₂ gradients for non-pond wind directions are significantly larger than those for pond directions, and the signal-to-noise ratio of the CO₂ flux significantly lower than for CH₄. Regardless, we have now added the following paragraph to the paper in section 3.2:

"It is possible to calculate $K_c$ values based on CO₂, in order to avoid potential circularity arguments when calculating gradient fluxes of CH₄ using this approach. However, the CO₂ flux signal from this pond was confounded by the strong natural variability of the CO₂ background, and the smaller signal-to-noise ratio of the pond CO₂ flux compared to the CH₄ flux (Fig. S1). Regardless, $K_c$ values based on CO₂ were calculated, and found to be noisier but statistically not different from those based on CH₄ (t-test p-value= 0.09, based on fluxes binned into 16 wind direction sectors). It would also be possible to base the calculated $K_c$ values on the sensible heat flux instead of the momentum flux, but due to the absence of significant heat fluxes at night, this would not provide the continuity that the momentum fluxes afford."

4) Last but not least, many of my previous comments were either ignored (e.g. text is not well organized: in methods section EC and gradient instrumentation descriptions are not well separated, in the results on-pond and off-pond results are mixed and the reader easily gets lost, spectral corrections are still ignored, gradient-EC (in)dependence still not discussed...), not applied everywere in the manuscript (for example eddy covariance → EC, including medians together with means, some clarifications, e.g. about tube dimension), or were answered in the author response, but not applied in the revised manuscript (e.g. discussion on shifting winds affecting gradient fluxes are also affecting EC fluxes, medians not included in the text even when they were in the response). Many of the comments listed below are the same as before, as they were not implemented in the revised manuscript. Overall, it seems not much attention was given to the revision of the manuscript (e.g. a figure is referred that does not exist).

Response: We appreciate the reviewer pointed out these points potentially causing confusion. We did not understand the point about including median together with means in the first review since there was some confusion about line numbers. We have now put the median flux of flux chamber results in our revision. All the points mentioned above were addressed individually in the following detailed comments and in the revised manuscript.

Detailed comments (line numbers refer to line numbers in the "track changes" version of the manuscript):

Table 2: What are BDL and NA (mention in caption)

Response: fixed.

Fig 2: Write the correlation coefficient in the plot.

Response: fixed.

Fig 3: What I meant by my previous comment, was that the bins are not of equal size in terms of number of data points included in each bin. And that is why the fit does not really follow the original datapoints. How does the fit change, if you use equal number of datapoints for each bin? What is the fit equation?

Response: There was an error in the figure caption of Figure 3. The yellow points are the Sc of the entire data, including pond directions and non-pond directions. The boxes and fittings were on Sc of pond direction only. To make this clearer, the figure caption is revised, and in the revised Figure 3 the Sc points when the wind was from the pond direction are highlighted in black. Please note, since EC flux changed after the spectral correction, the calculated Sc values for each half-hour also changed, so the data in the revised Figure 3 are different than the original points. To reflect the change, the green line showing the original fitting with the original data points are included in the figure below for reviewers to see.

We also accepted reviewer's suggestion to bin data into bins with an equal number of points, and performed the same fitting with the median of each 10-point bin. The medians of each bin are shown as blue circle (z/L is the median z/L of each bin), and the new fit is the blue solid curve. In addition, we also tried our original way of binning data, and fitting results are shown by a red solid line. Fitting results of these two binning approach are close. We have accepted reviewer's suggestion, and only results from the approach of binning every 10 points (blue line) is used in the final results in the revised manuscript.

The equation of the new fit:

$$S_c = \begin{cases} 0.08 + 3.13 \times 10^{-9} e^{\left(\frac{\frac{z}{L}+19.5}{1.008}\right)}, & \frac{z}{L} < -0.18 \\ 0.74, & \frac{z}{L} \geq -0.18 \end{cases}$$

[Figure]

Fig 5: I still advice to use a uniform color scheme here, as the present one is highlighting differences in the range of 1.9-3 ppm, which are not really as dramatic as the ones from 1.9-9 ppm. My suggestion is to
use a uniform color scheme, e.g. similar colors as you have used in scales 3-9 ppm. I recommend to read a recent Nature Communications article about colormap choises (https://doi.org/10.1038/s41467-020-19160-7). Adding the radius lines (0.2, 0.4, 0.6, 0.8, 1) on top of the wind rose would help the reader a lot (same goes for Fig S1).

Response: fixed. Please note, the original Figure S1 has become Figure S2.

Fig 6: The Xtick labels are too close to each other. Widen their distance or make the labels e.g. in 45 degree angle. You can also leave out the hours, just put dates in each xtick and specify in the caption that date tick represents midnight. Why is the lowest panel missing horizontal grids, when other panels have
them?
Response: fixed.

Fig 7: The shaded areas are not the same that you mention in the text!

Response: thank you for catching this. The shades have been modified to only include wind from the pond sectors accounting for a safety margin, i.e. $WD \geq 286°$ or $WD \leq 76°$. The original shades included sectors covering the shorelines.

Fig S5: Somehow mark in the figure daytime/nighttime.

Response: fixed. The sunrise and sunset timing are labeled.

Add the footprints of stable/unstable/neutral conditions to the supplement.

Response: This has been inserted into the supplement as Figure S3, and mentioned in the revised manuscript.

L18-20: Larger footprint together with frequent sampling.

Response: We modified this sentence as: "These results indicate that the larger footprint together with high temporal resolution of micrometeorological methods results in more robust emission estimates representing the whole pond."

L29: oil sands

Response: fixed.

L56: Not only emissions, but also uptake, can be measured with the technique. Change to "gas fluxes" or "surface-atmosphere exchange"

Response: fixed.

L59-60: The sentence still reads that the fluxes are well defined spatially. Reformulate the sentence.

Response: We have reformulated this sentence:
"Tailings ponds represent a useful testing ground for a multi-method comparison of flux measurement techniques due to their reliability as sources of significant fluxes, relatively well-defined sources areas, and minimal other anthropogenic sources in the immediate vicinity."

L74: 1/2" inner or outer diameter?

Response: fixed. "outer diameter" has been inserted in the text.

L77: H2O not defined. Mention that these are used for EC measurements

Response: fixed.

L78: Even with the numbers and formula you gave in the response, I get the Reynolds number ~1600. I don't understand how you get 4500.

Response: The Reynolds number is indeed only around 1600. In light of this, we have removed the sentence about plug flow. We apologize for this mistake, which we cannot reproduce now, and thank the reviewer again for her persistence. Upon revisiting the flow and resulting delay times, we discovered that our settings for the automated delay calculation (through cross-covariance peak detection) in EddyPro were incorrect and resulted in incorrect delay times, affecting 54% of all data points. Fixing this increased the covariances on average by about 15%. Adding this to the spectral corrections explained above yielded an overall average increase in the fluxes by about 28% for $CH_4$, and 35% for $CO_2$.

L73-78: Lots of confusion about sampling tube lengths. In the original manuscript gas concentrations were measured through 45 m tube and EC through 40 m tube. In the response you mention 40 m for gradient and 30 m for EC, in the revised manuscript gradients are measured with 45m and EC with 30 m. This should be quite straightforward and well documented...

Response: Sorry about the confusion. The eddy covariance tube was definitely 30m long, and all the gradient tubes were 40m in length.

L89: "Friction velocity ($u*$) can also be calculated from measured u, v, and w.". There is no point to this sentence if you don't present the equation, or tell how it is calculated.

Response: The equation of $u_*$ has been inserted there in the revised manuscript.

L116: EC has already been defined....

Response: fixed.

L120: "which in this study limits the method to sensible and latent heat ($H_2O$) fluxes, momentum, $CO_2$ and $CH_4$ fluxes." Since you don't show or describe other measurements, this sentence is unnecessary

Response: we indeed have sensible heat fluxes shown in Fig. 4 (e) and have used momentum fluxes in equation (4) and the $CO_2$ fluxes in Table 2 and section 3.2 and 4.7, so we would like to retain this sentence.

L121: eddy covariace -> EC! Check this everywhere in the manuscript!

Response: All the "eddy covariance" have now been replaced with "EC".

L121: replace "..the eddy covariance method simply calculates the flux by averaging.." by "in the EC method, flux is calculated by averaging…"

Response: fixed.

L126-130: Mention that you assume linear concentration profile for storage change flux calculation

Response: implemented.

L132: Mention the average time lag in the text.

Response: implemented.

L130-135: Mention that each wind sector was processed individually and how you took into account the different roughness elements of different wind sectors.

Response: we inserted this detail at the end of Section 3.1.

L190: Again, shifting wind directions are also a problem for EC, not just gradient fluxes! So these occasions are not probably due to shifting wind!

Response: Correct, they would also be a problem for EC. We have eliminated this sentence, since upon re-examination it contained circular reasoning and didn't actually describe what we did. A speculative reason for such reversed gradients would be a surface layer out of equilibrium with the pond, with elevated $CH_4$ concentrations being observed at 32m relative to those at 8m, since the concentration footprint at 32m is significantly larger than that at 8m. As we mention in the last paragraph of 4.2, the gradient footprint is equivalent to the eddy covariance footprint at the geometric mean height of the two gradient levels, but the underlying assumption is an infinitely homogeneous fetch. If the 32m concentration footprint reaches beyond the pond limit, this assumption may be broken.
In our revision, we slightly modified our process, and kept some of the negative gradient fluxes. Due to the scatter natural of the half-hour gradient fluxes, we excluded outliers of the lowest 2.5% and the highest 2.5% of data. In this way, we did not just eliminate negative gradient fluxes. We have stated this process clearly in the Section 3.2: "To lessen the impact of extreme outliers, the final pond average fluxes reported were based on gradient fluxes between the $2.5^{th}$ and $97.5^{th}$ percentiles".

L200-203: Mention the units you have used!

Response: implemented.

L218: I looked the report cited here and still did not find how the chamber fluxes are calculated. Please provide a formula for flux calculation. I understand this may not be the focus of this paper, but chamber fluxes are quite sensible to the calculation method used, and thus I think it is valuable information for the reader.

Response: Sorry, this time we understand what was being asked. The report of Alberta Environment and Parks Section 3.6 has included the EPA method. To implement this comment, we included the EPA (1986) report and cited the key equation (3-5) in the revised manuscript as equation (8).

https://nepis.epa.gov/Exe/ZyNET.exe/930013RX.TXT?ZyActionD=ZyDocument&Client=EPA&Index=
1986+Thru+1990&Docs=&Query=&Time=&EndTime=&SearchMethod=1&TocRestrict=n&Toc=&Toc
Entry=&QField=&QFieldYear=&QFieldMonth=&QFieldDay=&IntQFieldOp=0&ExtQFieldOp=0&Xml
Query=&File=D%3A%5Czyfiles%5CIndex%20Data%5C86thru90%5CTxt%5C00000029%5C930013R

X.txt&User=ANONYMOUS&Password=anonymous&SortMethod=h%7C-
&MaximumDocuments=1&FuzzyDegree=0&ImageQuality=r75g8/r75g8/x150y150g16/i425&Display=h
pfr&DefSeekPage=x&SearchBack=ZyActionL&Back=ZyActionS&BackDesc=Results%20page&Maxim
umPages=1&ZyEntry=1&SeekPage=x&ZyPURL

L219: N2O not defined

Response: fixed.

L220-223: Again, I still find this section quite confusing. Is this then the final flux you use in the flux comparison, or something else? Which flux measurements are used for this? EC, gradient, chamber, IDM? Provide more details.

Response: Yes, these are the final average fluxes representing the pond emission during this study for
each of the methods (EC, gradient, and IDM). Section 3.5 has been modified: "The area weighted averages of fluxes results are summarized in Table 1 and serve as the final average fluxes representing the whole pond over the study period."

L303-308: Again, it is not clear wether you are now describing the off-pond or pond fluxes. For pond fluxes, wind should play at least some role, enhancing the turbulent transport of gases from the pond. Specify in the text if you are focusing on pond or off-pond fluxes in this discussion, since in the previous paragraphs you are describing both. As this indeed is from pond direction only (as you state in the response), discuss why they should not play any role, i.e., why more mixing would not bring up methane
produced deeper in the pond. Looking at Fig. S3, even though there is no clear linear relationship, it is still clear that lowest fluxes are not measured at high wind speed and highest fluxes are not measured at lowest wind speeds. So there is some kind of relation to wind.

Response: The r of the linear regression of EC flux (from the pond direction) and wind speed at 8m is 0.4,
so we modified the text: *"Relationships between the flux when the wind was from the pond and various meteorological parameters were investigated, and results show that fluxes showed weak dependence of wind speed, $u_*$, water surface temperature, or the temperature difference between the water surface and 8 m (Fig. S5)."*
It is of course physically possible that higher wind speeds could enhance mixing of the water near the
water-air interface, as has been observed and parameterized elsewhere **(cf. Cole, J. J. and Caraco, N. F.: Atmospheric exchange of carbon dioxide in a low-wind oligotrophic lake measured by the addition of $SF_6$, Limnol. Oceanogr., 43, 647–656, 1998)**. Visually we saw little wave formation even during windy periods, suggesting that the chemical composition, and possibly the presence of surface films, suppressed transfer of momentum from the air to the water. Also, it is unlikely that the production rate of methane by
microbes in the lower (anoxic) strata of the pond are affected on a short (< daily) time scale; in other words, the source strength is likely mostly independent of wind speed, even if the transport mechanism varies. We modified the text as follows:
*"Relationships between the flux, when the wind was from the pond, and various meteorological parameters were investigated, and results show that fluxes showed* weak *dependence of wind speed, $u_*$, water surface*
*temperature, or the temperature difference between the water surface and 8 m (Fig. S4), i.e. they were not* major *drivers of the $CH_4$ emission rate. $CH_4$ at this site is mainly produced through the methanogenesis of hydrocarbon by the microbes in the fine tailings covering a range of depth in the pond (Penner and Foght, 2010; Siddique et al., 2011;Siddique et al., 2012), and therefore is not* directly *affected much by the meteorological conditions at the surface or above the pond."*

 L324: Report also the median flux differences, as requested before.

Response: implemented.

L327-338: You are still not discussing the relation to EC fluxes. Yes, the Kc was determined from a fit made to EC fluxes, and yes, almost all gradient flux methods require some input of the EC system. But how would the results look if you determine Kc from CO2 flux instead of CH4 flux? That would lead to at least a bit more independent comparison to EC CH4 flux. By minimum, you should discuss how the derivation of Kc from EC CH4 flux is affecting your comparison, or justify why it is not affecting at all.

Response: Please see our response to major point 3) above. Calculating $K_c$ from the $CO_2$ data produced noisier results for the reasons discussed above, but the results were statistically not significantly different from those for $CH_4$. In other words, using the $K_c$ derived from $CO_2$ would give us similar gradient fluxes, but they would be even noisier than those from $CH_4$. Dividing the eddy flux by the gradient (measured by a separate instrument, incidentally) results in a diffusivity that represents the transport of a nonreactive gas in general; we chose the best-resolved gas (in terms of signal-to-noise) to calculate it, and that happens to be $CH_4$ in this location. Our approach then hinges $K_c$ on the stability ($z/L$), thereby removing it another step from direct correlation with the EC flux used to calculate the $K_c = f(Sc)$ relationship. The magnitude of the gradient flux is a direct function of the EC flux, thereby ensuring that the gradient flux will on average be about the same as the EC flux. An independent verification of the approach is given by our $Sc$ values being similar to the few numbers in the published literature.

L335: Which studies? Add references to the sentence.

Response: we meant studies discussed in the sentences following that sentence. A study with another tailings pond (Zhang et al. 2019), and studies with other water surfaces (Schubert et al. 2012; Podgrajsek et al. 2014; Erkkilä et al. 2018). In the revised manuscript, we have inserted those references at the end of this sentence.

L340: Again, you cannot really say they are lower when the medians are almost the same, and also meand within the confidence intervals. It would be good to try some statistical tests to see if they really differ or not. Add it to the discussion.

Response: A t-test was performed with half-hour gradient fluxes with the both the variable and constant

Sc approaches and the result was p $<10^{-21}$, indicating the two results are indeed statistically different. The pond median flux with the constant Sc approach was 34% lower than the median of fluxes with the variable Sc approach, and the mean was 33% lower. We have implemented this comparison in the main text in the last paragraph in Section 4.4: "Gradient flux calculated from a constant $S_c$ were significantly lower than gradient fluxes with the variable $S_c$ approach (p $< 10^{-21}$, pond average mean/median is

33%/34% lower)."

L351: There is no Fig 9... You might mean Fig. 8b.
Response: Correct, thank you for catching this. Also, we have decided to move Fig. 8 into the supplement as Figure S7.

L366: Again about the bubbling zones.
Response: "The locations of the 15 flux chamber measurements are marked in the revised Figure 1. As can be seen, most of them fall within the EC footprint. It is possible that the sudden bursts of CH4 could invalidate the flux chamber calculation and lead to an underestimation of flux, as discussed in Zhang et al.

(2019). We wrote this in line 334-337. Integrated over the footprint of micrometeorological flux measurements, the intermittent nature of ebullition will have a minimal effect."
I did not find any discussion on lines 334-337 on the subject (on the previous manuscript, on the revised manuscript, or the author response).

Response: This might be the line number issue. We have had this discussion in all the versions since the original draft. In the middle of Section 4.6 when we first discussed Zhang et al. (2019), we have "Zhang et al. (2019) measured $CH_4$ emission from another tailings pond, and reported flux chamber measurements were more than 10 times greater than fluxes from the EC method. They stated that strong eruptions of bubbles could overwhelm the chamber to result in a local underestimation of the flux. On the other hand, the lower EC flux estimate suggests that the area average flux was being overestimated by extrapolation from the chambers, which may have preferentially been located over bubble zones. Their EC fluxes were two orders of magnitude smaller than $CH_4$ flux in this study. Results from this study and Zhang et al. (2019) suggest that average tailings pond $CH_4$ emission extrapolated from a few individual flux chamber measurements may significantly underestimate or overestimate fluxes relative to area-averaging micrometeorological measurements."

L405: EC

Response: fixed.

L416: As a side note, Pond 2/3 is a great name for a pond!

Response: There are historical reasons for the name. There used to be adjacent ponds numbered "2" and "3", but they were merged in the 1980s and to retain this history in the name, Suncor labelled the new unified pond "2/3".

L459: Abbreviation CO2eq is (still) not defined

Response: fixed. "($F\_CO_{2eq}$)" is inserted into that sentence after "the equivalent $CO_2$ flux"

Somewhere in the discussion, mention how high these CH4 emissions are compared to natural (wetland/lake/pond) emissions from other studies, just to give some idea of the flux magnitude.

Response: we have inserted the following at the end of the first paragraph in Section 4.7.

"Natural lakes and wetlands emit at rates typically on the order of 0.005-0.05 g m$^{-2}$ d$^{-1}$ (Sanches et al., 2019)."

L462: Mention the methods used.

Response: implemented. We have modified this sentence:

"Results in this study have provided several estimates of the emission of $CH_4$ from this tailings pond using EC, gradient, and IDM methods,…"

L466: "in 2017" is not needed here

Response: deleted.

L467-468: "micrometerological flux measurements"

Response: fixed.

L468: "larger footprint together with high temporal resolution"

Response: fixed.

L469: To be accurate, the measurements are still not representing the whole pond. But they are representing most of the pond area. Reformulate the sentence.

Response: That is true. The modified sentence states: "The better agreement between the three
micrometeorological measurements flux results suggests that the larger footprint of micrometeorological measurements results in more robust emission estimates representing most of the pond area."

L471: Further studies of what? EC, chamber, IDM, gradient, temperature, wind, what? Flux measurements in general? At what time resolution?

Response: the last sentence has been modified: "To investigate seasonal patterns, further studies measuring $CH_4$ fluxes using micrometeorological methods at this pond or other tailings ponds during other times of the year are recommended."

---

## Author Response (AR3)

**You et al: Methane emissions from an oil sands tailings pond: A quantitative comparison of fluxes derived by different methods**

**amt-2020-116**

20 January 2021

**Response to reviewer's comment:**
Reviewer's comments are in black, and authors' response are in blue.

The manuscript has improved considerably since the last version. The authors have implemented statistical tests to their flux comparisons, which makes the results and conclusions of this paper stronger. Figures and their captions have been improved and made more clear. Some mistakes/errors in flux processing have been found and fixed. I only have few minor points to correct before publication.

(1) I am still missing discussion on how the gradient flux method used in this study would benefit future flux measurements. If this gradient flux method relies on the EC setup, why is the gradient flux needed? In general, if one has an EC system running gas flux measurements, the gradient flux would not be needed because it is already directly measured with EC. Thus, gradient flux measurements are generally used in places where an EC setup is not possible. Can the results and methods of this study be applied to future gradient flux measurements? If so, when, where and how? For example, is the Sc function you have given in Eq. 6 applicable to future pond studies anywhere, or could it be used at the same site for gradient fluxes if the EC system is taken down? Add a couple of sentences about the wider use and application of this method to discussion.

Response: The objectives of the project were much broader than just methane fluxes; we quantified fluxes for 68 VOCs, 23 PACs, 12 organic acids, 7 reduced sulfur compounds and ammonia using gradient flux and inverse dispersion methods. These results will be disseminated in upcoming publications. For almost all of these gases, eddy covariance was not an option since the existing or available instrumentation did not have the required time resolution. Therefore we wanted to make sure that our gradient and inverse results agreed with the eddy covariance results, and designed the study to allow for a cross-comparison of all methods based on at least one gas with a clear, strong flux signal, and in this project that was methane.
Yes, the methods shown in this study can definitely be applied to future gradient flux measurements. We completely agree that if a flux can be measured using EC, then that is the preferred method, which is why we used it as our "benchmark". For most gases, that is not possible with current technology, and that is where gradient or inverse dispersion methods become necessary.
We see no reason why the Schmidt number relationship we derived would not be applicable quite broadly, since it is primarily a function of the characteristics of atmospheric turbulence. There may be dependencies that our comparatively large but still quite limited dataset cannot tease out, and for those, additional studies over different surface types and broader stability

scenarios would be required. Since the literature on atmospheric Schmidt numbers is very limited, even our average Sc estimate will be a useful starting point for future gradient $CH_4$ flux studies.

We have revised the manuscript as follows:
   a) The last sentence of the Introduction has been modified to more broadly reflect the applicability of this comparison study: "This manuscript describes the results of a comparison of flux chambers, EC, gradient and IDM approaches for estimating emission rates of $CH_4$, to verify the suitability of these methods for quantifying fugitive emissions from such sources."
   b) In the Discussion Section 4.4., line 320, we modified one sentence to: "The gradient fluxes of $CH_4$ agreed well with EC flux in our study, providing a basis for applying the derived Kc values to calculate gradient fluxes for a variety of other gases emitted by the pond (e.g. You et al. (2021))."
   c) Also in Section 4.4. a sentence was added at the end stating "While the function derived (Eq. (6)) is primarily a function of the characteristics of atmospheric turbulence and should have broad applicability, it is based on a limited data set and should be verified in other settings in future studies."

(2) Sometimes the results of the t-test are presented as "p-value=X" and sometimes as "p<Y". Make it consistent.
Response: Thank you for pointing this out. We have corrected them to be consistent as "p" only.

Detailed comments:

L18-19: Suggest to reformulate as "The results show that the larger footprint together with high temporal resolution of micrometeorological flux measurement methods may result in more robust estimates of the pond greenhouse gas emissions." since the whole pond is actually not measured and this study is about greenhouse gas emissions, not e.g. particle emissions.
Response: Accepted and implemented.

L75: eddy covariance = EC
Response: Replaced.

L112: "...from their means"
Response: Revised.

L231-232: This is stated already earlier in the text, suggest to remove.
Response: Removed.

L302: Replace "less" with "lower"
Response: Replaced.

L342: Mention that these are the chamber fluxes (although evident from the title). It comes as a surprise here that the measurements are done in a bubbling zone. Please mention it already in e.g. methods section when describing chambers, or in introduction.

Response: Chambers were not placed in bubble zones exclusively. We have modified the relevant statements to "in and around bubbling zones" in Section 3.4 line 212, and Section 4.6 line 344.

Sect 4.6: Chambers may indeed overestimate/underestimate a long-time emission estimate (like a yearly GHG budget) due to the lack of spatial and temporal resolution, but they can also be most useful in estimating smaller source areas (e.g. close to shore) and shorter timeframe emissions, especially in remote locations. Add a sentence or two about this in the discussion. At the moment it sounds like you are saying that chambers are not useful at all.

Response: We changed the last sentence of 4.6 to "In conclusion, while flux chambers present advantages in terms of finer spatial and temporal resolution for small sources or locations with high spatial heterogeneity, reliance on a limited number of flux chamber measurements can result in significant year-to-year variability, and spatially integrating methods such as eddy covariance or gradient fluxes will generally provide more representative averages."

L422: Global warming potential is already defined in Introduction. Define the acronym GWP in Introduction and use it here.

Response: Implemented in line 39 and line 426.

L437: Suggest to replace "during other times of the year" with "throughout the year"

Response: Replaced.